# Cestode larvae excite host neuronal circuits via glutamatergic signalling

Anja de Lange[1,2,3†], Hayley Tomes[1,2,3†], Joshua S Selfe[1,2,3],
Ulrich Fabien Prodjinotho[4], Matthijs B Verhoog[1,2], Siddhartha Mahanty[5],
Katherine Ann Smith[3,6], William Horsnell[3,7,8], Chummy Sikasunge[9],
Clarissa Prazeres da Costa[4], Joseph V Raimondo[1,2,3]*

[1]Division of Cell Biology, Department of Human Biology, Faculty of Health Sciences, University of Cape Town, Cape Town, South Africa; [2]Neuroscience Institute, Faculty of Health Sciences, University of Cape Town, Cape Town, South Africa; [3]Institute of Infectious Disease and Molecular Medicine, Faculty of Health Sciences, University of Cape Town, Cape Town, South Africa; [4]Center for Global Health, TUM School of Medicine, Technical University of Munich, Munich, Germany; [5]Department of Medicine, The Peter Doherty Institute for Infection and Immunity and the Victorian Infectious Diseases Service, University of Melbourne, Melbourne, Australia; [6]School of Biosciences, Cardiff University, Cardiff, United Kingdom; [7]Institute of Microbiology and Infection, University of Birmingham, Birmingham, United Kingdom; [8]Laboratory of Experimental and Molecular Immunology and Neurogenetics (INEM), UMR 7355 CNRS-University of Orleans, Orleans, France; [9]School of Veterinary Medicine, Department of Paraclinicals, University of Zambia, Lusaka, Zambia

*For correspondence:
joseph.raimondo@uct.ac.za

[†]These authors contributed equally to this work

Competing interest: The authors declare that no competing interests exist.

## eLife Assessment

This manuscript addresses infections of the parasite Taenia solium, which causes neurocysticercosis (NCC). NCC is a common parasitic infection that leads to severe neurological problems. It is a major cause of epilepsy, but little is known about how the infection causes epilepsy. The authors used neuronal recordings, imaging of calcium transients in neurons, and glutamate-sensing fluorescent reporters. A strength of the paper is the use of both rodent and human preparations. The results provide **convincing** evidence that the larvae secrete glutamate and this depolarizes neurons. Although it is still uncertain exactly how epilepsy is triggered, the results suggest that glutamate release contributes. Therefore, the paper is a **fundamental** step towards understanding how Taenia solium infection leads to epilepsy.

**Abstract** Neurocysticercosis (NCC) is caused by infection of the brain by larvae of the parasitic cestode Taenia solium. It is the most prevalent parasitic infection of the central nervous system and one of the leading causes of adult-acquired epilepsy worldwide. However, little is known about how cestode larvae affect neurons directly. To address this, we used whole-cell patch-clamp electrophysiology and calcium imaging in rodent and human brain slices to identify direct effects of cestode larval products on neuronal activity. We found that both whole cyst homogenate and excretory/secretory products of cestode larvae have an acute excitatory effect on neurons, which can trigger seizure-like events in vitro. This effect was mediated by glutamate receptor activation but not by nicotinic acetylcholine receptors, acid-sensing ion channels, or Substance P. Glutamate-sensing fluorescent reporters (iGluSnFR) and amino acid assays revealed that the larval homogenate of the cestodes Taenia crassiceps and Taenia solium contained high concentrations of the amino acids glutamate and aspartate. Furthermore, we found that larvae of both species consistently produce

and release these excitatory amino acids into their immediate environment. Our findings suggest that perturbations in glutamatergic signalling may play a role in seizure generation in NCC.

## Introduction

Neurocysticercosis (NCC) is the most prevalent parasitic infection of the central nervous system (CNS) (*Diop et al., 2003*; *Preux and Druet-Cabanac, 2005*). It is caused by the presence of larvae of the cestode *Taenia solium* in the brain (*Trevisan et al., 2016*). The most common symptom of NCC is recurrent seizures (*Garcia et al., 2017*). As a result, NCC is one of the leading causes of adult-acquired epilepsy worldwide (*Nash et al., 2013*), resulting in significant morbidity and mortality. In *T. solium* endemic areas, 29% of people with epilepsy also had NCC (*Ndimubanzi et al., 2010*). Despite the impact of NCC, there is a paucity of studies investigating the seizure mechanisms involved (*de Lange et al., 2019*). As far as we are aware, there have been no studies to date that investigated the direct, acute effects of cestode larval products on neurons. As a result, precisely how larvae perturb neuronal circuits is still relatively poorly understood.

In NCC seizures may occur at any stage following initial infection (*Garcia et al., 2017*). It is thought that inflammatory processes in the brain play an important role in the development of recurrent seizures (*Vezzani et al., 2016*). Previous research investigating seizure development in NCC has therefore typically focused on how the host neuroinflammatory response to larvae may precipitate seizures (*de Lange et al., 2019*; *Nash et al., 2015*). One study by Robinson et al. found that production of the inflammatory molecule and neurotransmitter, Substance P, produced by peritoneal larval granulomas, precipitated acute seizures (*Robinson et al., 2012*). In addition to host-derived substances, cestodes themselves are known to excrete or secrete various products that interact with host cells in their vicinity. Cestode larvae-derived factors are known to modulate the activation status of immunocytes such as microglia and dendritic cells (*Sun et al., 2014*; *Vendelova et al., 2016*). However, comparatively little is known about how factors contained in, or secreted by, cestode larvae might affect neurons and neuronal networks directly, including whether these may have pro-seizure effects.

To address this, we used whole-cell patch-clamp recordings and calcium imaging in rodent hippocampal organotypic slice cultures and in acute human cortical brain slices to demonstrate the direct effects of larval products on neuronal activity. We find that both the whole cyst homogenate and the excretory/secretory (E/S) products of *Taenia crassiceps* and *Taenia solium* larvae have strong, acute excitatory effects on neurons. This was sufficient to trigger seizure-like events (SLEs) *in vitro*. Underlying SLE induction was larval-induced neuronal depolarization, which was mediated by glutamate receptor activation and not by nicotinic acetylcholine receptors, acid-sensing ion channels (ASICs), or Substance P. Both imaging using glutamate-sensing fluorescent reporters (iGluSnFR) and direct measurements using amino acid assays revealed that the homogenate and E/S products of both *T. crassiceps* and *T. solium* larvae contain high levels of the excitatory amino acid glutamate and, to a lesser extent, aspartate. Lastly, we provide evidence that larvae of both species, to varying degrees, can produce and release these excitatory amino acids into their immediate environment. This suggests that these parasites release amino acids that could contribute to seizure generation in NCC.

## Results

### *T. crassiceps* homogenate excites neurons and can elicit epileptiform activity

To investigate the potential acute effects of *T. crassiceps* larvae on neurons, *T. crassiceps* larval somatic homogenate was prepared using larvae, harvested from the peritonea of mice, which were freeze-thawed and homogenized (see Materials and methods and *Figure 1A*). Whole-cell patch-clamp recordings were made from CA3 pyramidal neurons in rodent hippocampal organotypic brain slice cultures and layer 2/3 pyramidal neurons in human cortical acute brain slices. Pico-litre volumes of *T. crassiceps* homogenate were directly applied to the soma of neurons using a custom-built pressure ejection system (*Figure 1A*; *Forman et al., 2017*). Application of the homogenate (20-ms puff) elicited immediate, transient depolarization of the membrane voltage in recordings from rat, mouse, and human neurons (*Figure 1B–D*, *Figure 1—source data 1*). Increasing the amount of homogenate

**eLife digest** One of the main causes of epilepsy in adults – particularly in developing countries – is a parasitic brain infection called neurocysticercosis. This can happen when people swallow tapeworm eggs, which hatch into larvae and migrate throughout the body. When these larvae infect the brain, they form structures called cysts, which can cause seizures.

It is thought that inflammation in the brain contributes to the development of seizures in neurocysticercosis, but how this might work is still poorly understood. The larvae produce chemicals that can interact with nearby cells in the body, including the defensive cells of our immune system. However, it remains unknown whether those chemicals also interact with brain cells.

De Lange, Tomes et al. set out to determine if tapeworm larvae produced any specific chemicals that affect the activity of brain cells, and if they might play a role in epileptic seizures. To do this, the researchers collected materials from tapeworm larvae, which included both the substances they naturally released and a mixture made from crushed whole larvae. They then applied these substances to brain tissue grown in cell culture while recording the electrical activity of individual brain cells.

Experiments using brain tissue derived from rats, mice and humans revealed that the larval products made brain cells more excited and led to them firing more electrical signals than normal. This excitation was strong enough to trigger larger patterns of activity across the brain tissue that mimicked the effect of an epileptic seizure.

Further biochemical analysis of the larval products and the larvae themselves revealed that tapeworm larvae continuously release a chemical called glutamate, which is known to excite brain cells. These results suggested that tapeworm larvae might cause epilepsy by producing excess glutamate and overexciting brain cells – a mechanism similar to the way that other brain conditions, like tumors, also trigger seizures.

This work has revealed a new mechanism for how tapeworm larvae might cause seizures in neurocysticercosis. The next step will be understanding how the larvae release glutamate into the brain, for example, if they actively produce it, or if it is passively released when they die. In the future, de Lange et al. hope this knowledge will help develop new treatments that help prevent seizures in people with neurocysticercosis.

---

delivered by increasing the pressure applied to the ejection system resulted in increasingly large membrane depolarization, which could trigger single or multiple action potentials (*Figure 1B–D*). To further explore the acute excitatory effect of *T. crassiceps* on neurons, neuronal networks, and the propagation of network activity, we performed fluorescence $Ca^{2+}$ imaging in mouse hippocampal organotypic brain slice cultures. Neurons were virally transfected with the genetically encoded $Ca^{2+}$ reporter, GCAMP6s, under the synapsin promoter and imaged using widefield epifluorescence microscopy (*Figure 1E*, *Figure 1—video 1*, and Materials and methods). To simulate a pro-ictal environment, a low $Mg^{2+}$ artificial cerebrospinal fluid (aCSF) was used (0.5 mM $Mg^{2+}$) and neurons in the dentate gyrus were imaged whilst small, spatially restricted puffs of *T. crassiceps* homogenate were delivered every 15 s using a glass pipette (*Figure 1E*). The cells within the direct vicinity of the puffing pipette showed a sharp increase in fluorescence immediately following the delivery of *T. crassiceps* homogenate (*Figure 1F*, $t_1$) for all three puffs, indicating $Ca^{2+}$ entry following membrane depolarization and action potential generation. Cells in the periphery showed notable increases in fluorescence at a delayed interval following some (but not all) puffs (*Figure 1F*, $t_2$). The excitation of these cells are likely the result of being synaptically connected to the cells that were exposed to the puff itself. Indeed, a current-clamp recording from a neuron in the same slice (*Figure 1F*, inset) indicates that a single, spatially restricted puff of *T. crassiceps* homogenate can elicit the onset of a regenerative SLE lasting far longer than the puff itself.

Most cell types tend to have a high $[K^+]_i$, therefore it is conceivable that the *T. crassiceps* homogenate could have a high $K^+$ concentration, which could potentially account for its depolarizing and excitatory effects on neurons. To address this, we directly measured the ionic composition of the *T. crassiceps* homogenate using a Roche Cobas 6000. The $K^+$ concentration of the *T. crassiceps* homogenate was 11.4 mM ($N = 1$, homogenate prepared from >100 larvae) as compared to 3.0 mM in our standard aCSF ($N = 1$). Direct application of 11.4 mM $K^+$ via puff pipette during whole-cell current-clamp

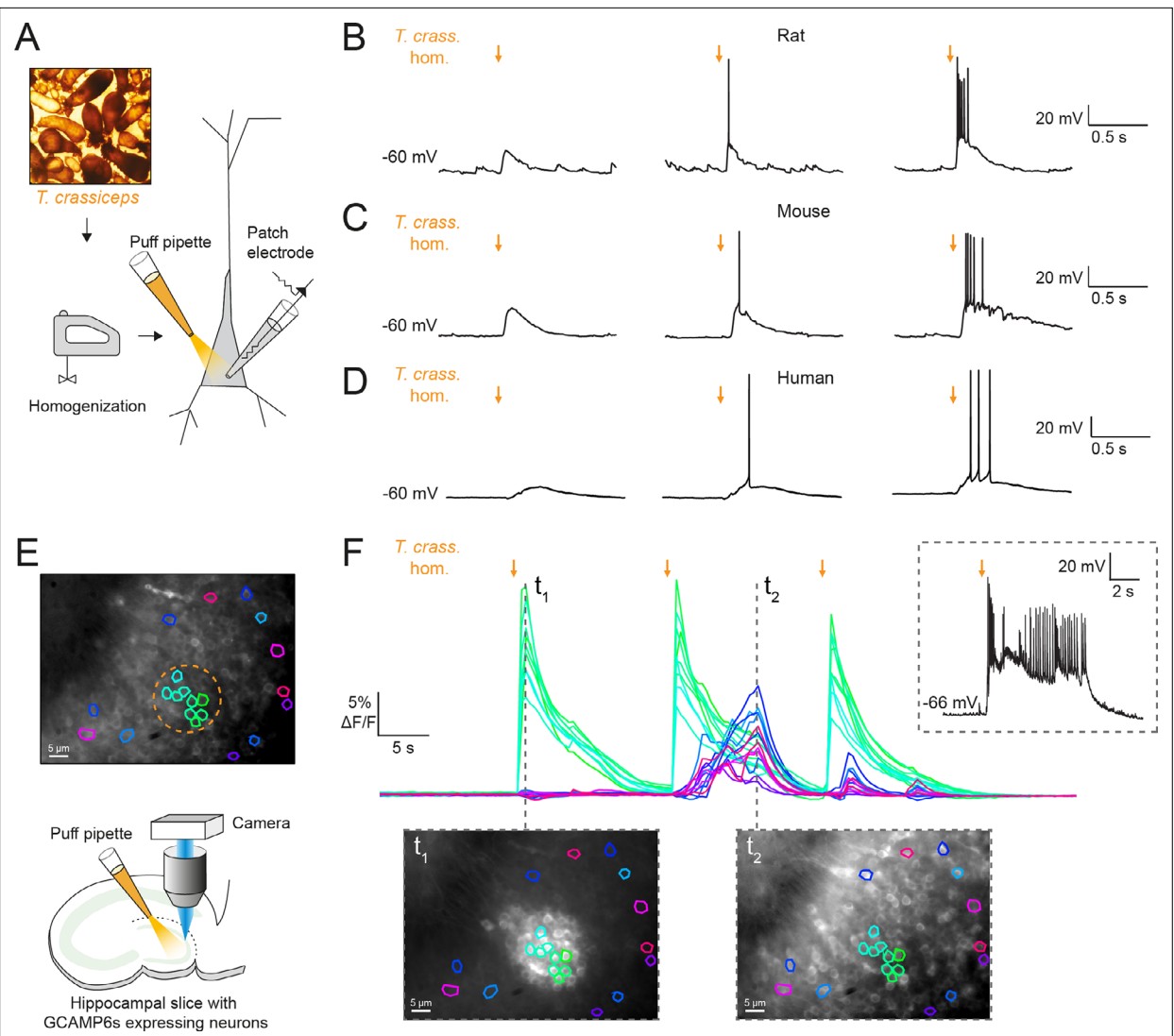

**Figure 1.** *Taenia crassiceps* homogenate excites neurons and can elicit epileptiform activity. (**A**) Schematic showing the experimental setup in which whole-cell patch-clamp recordings were made from CA3 hippocampal pyramidal neurons in rodent organotypic slice cultures whilst a puff pipette delivered pico-litre volumes of homogenized *Taenia crassiceps* larvae (*T. crass.* hom.) targeted at the cell soma. (**B**) Current-clamp recording from a rat pyramidal neuron whilst increasing amounts of *T. crass.* hom. was applied via the puff pipette (left to right, orange arrows). Small amounts of *T. crass.* hom. resulted in depolarization (left), increasing amounts by increasing the pressure applied to the ejection system elicited single (middle) or even bursts of action potentials (right). (**C**) As in 'B', identical effects of *T. crass.* hom. could be elicited in current-clamp recordings from a CA3 hippocampal pyramidal neuron in a mouse organotypic brain slice culture. (**D**) As in 'B' and 'C', identical effects of *T. crass.* hom. could be elicited in current-clamp recordings from a human frontal lobe layer 2/3 cortical pyramidal neuron in an acute brain slice. (**E**) Top: widefield fluorescence image of neurons in the dentate gyrus of a mouse hippocampal organotypic brain slice culture expressing the genetically encoded $Ca^{2+}$ reporter GCAMP6s under the synapsin promoter in artificial cerebrospinal fluid (aCSF) containing 0.5 mM $Mg^{2+}$. A subset of neurons used to generate the $Ca^{2+}$ traces in 'E' are indicated by different colours. The orange dotted circle indicates where *T. crass.* hom. was delivered using the puff pipette. Bottom: schematic of the experimental setup including puff pipette and CCD camera for $Ca^{2+}$ imaging using a 470-nm LED. (**F**) Top, d*F/F* traces representing $Ca^{2+}$ dynamics from the GCAMP6s expressing neurons labelled in 'E' concurrent with three puffs (30-ms duration) of *T. crass.* hom. 15 s apart. Bottom: two images of raw $Ca^{2+}$ fluorescence at two time points $t_1$ and $t_2$. Note how at time point $t_2$ neurons distant to the site of *T. crass.* hom. application are also activated, indicating spread of neuronal activity. Inset, top-right: current-clamp recording from a neuron in the region of *T. crass.* hom. application demonstrates seizure-like activity in response to *T. crass.* hom. application.

The online version of this article includes the following video, source data, and figure supplement(s) for figure 1:

**Source data 1.** Patch clamp cell properties and metadata.

**Figure supplement 1.** *Taenia crassiceps* homogenate depolarizes neurons, artificial cerebrospinal fluid (aCSF) does not.

**Figure 1—video 1.** *Taenia crassiceps* homogenate excites neurons and can elicit epileptiform activity.

https://elifesciences.org/articles/88174/figures#fig1video1

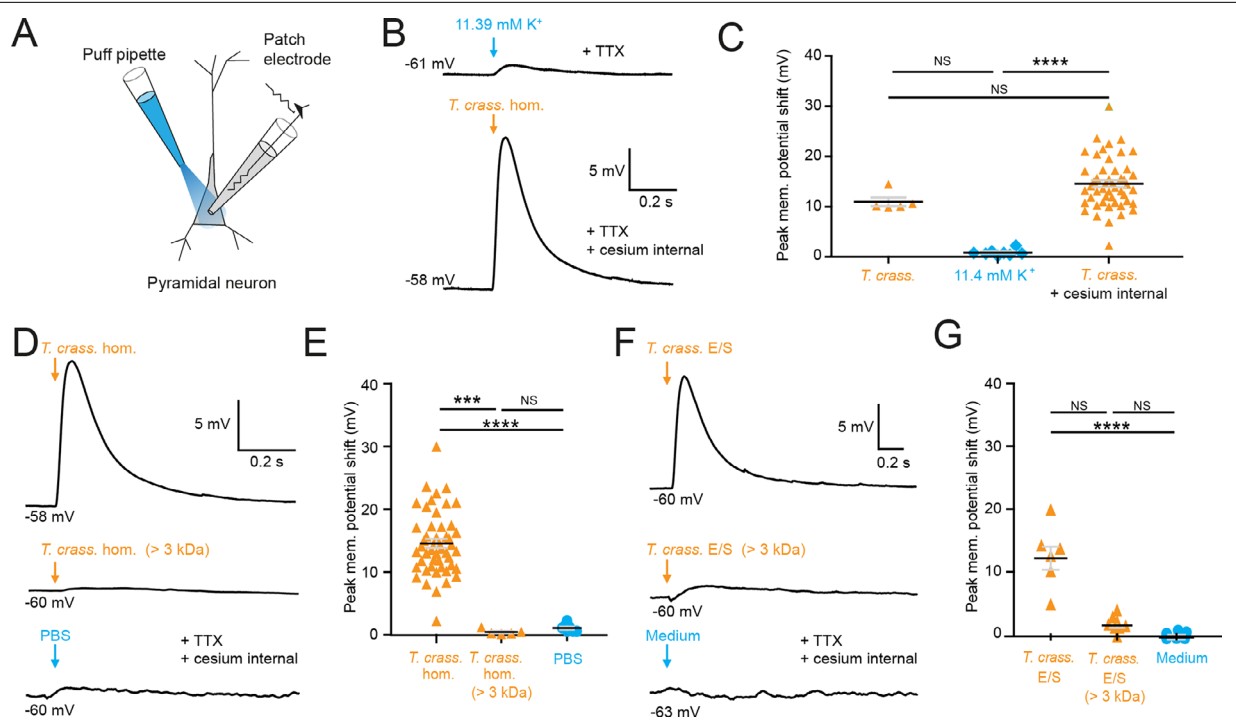

**Figure 2.** *Taenia crassiceps* homogenate and E/S product-induced neuronal depolarization is mediated by a small molecule. Schematic showing the experimental setup in which whole-cell patch-clamp recordings were made from CA3 hippocampal pyramidal neurons in rat organotypic slice cultures whilst a puff pipette delivered pico-litre volumes targeted at the cell soma. (**B**) Top trace: 20-ms puff of artificial cerebrospinal fluid (aCSF) containing 11.4 mM K⁺ (equivalent to the K⁺ concentration of *T. crassiceps* homogenate (*T. crass.* hom.)) caused modest depolarization. A standard internal solution was utilized, and 2 µM tetrodotoxin (TTX) was added to circulating aCSF to reduce synaptic noise in the voltage trace. Bottom trace: 20-ms puff of *T. crass.* hom. was applied, but a cesium-based internal solution was utilized to block K⁺ channels in the presence of TTX. Puffs of *T. crass* hom. resulted in sizeable depolarization. (**C**) Population data showing that responses to 11.4 mM K⁺ puffs were significantly smaller than that of the *T. crass* hom. when a cesium internal solution was utilized (*T. crass.* + cesium internal), but not when a standard internal solution was utilized (*T. crass.*). (**D**) Delivery of *T. crass.* hom. caused a depolarizing shift in membrane potential (top trace). The depolarizing response to *T. crass* hom. was largely abolished by dialysing out all molecules smaller than 3 kDa (middle trace). Phosphate-buffered saline (PBS), the solvent for *T. crass.* hom., did not induce a large neuronal depolarization (bottom trace). (**E**) Population data showing that the membrane depolarization induced by *T. crass.* hom is not due to the PBS solvent and is due to a molecule smaller than 3 kDa. (**F**) Delivery of a puff of *T. crass* excretory/secretory (E/S) products also caused a depolarizing shift in membrane potential (top trace), which was largely abolished by filtering out all molecules smaller than 3 kDa (middle trace). Culture medium, the solvent for the *T. crass* E/S did not induce depolarization (bottom trace). (**G**) Population data showing that the membrane depolarization induced by *T. crass.* E/S is not due to the culture medium solvent and is due to a molecule smaller than 3 kDa. Values with medians ± IQR; ***p ≤ 0.001, ****p ≤ 0.0001, NS = not significant.

The online version of this article includes the following source data for figure 2:

**Source data 1.** Patch clamp membrane voltages, cell properties, and metadata for all cells.

from rat CA3 pyramidal neurons resulted in a modest median positive shift of only 0.72 mV (IQR 0.51–1.04 mV, *N* = 8 from 4 slices, *Figure 2B, C*, *Figure 2—source data 1*) in the membrane potential, which was lower than the depolarization caused by puffs of *T. crassiceps* homogenate (median 10.12 mV, IQR 9.93–12.49 mV, *N* = 5 from 4 slices), although this did not reach statistical significance (p = 0.26, Kruskal–Wallis test with Dunn's multiple comparison test, *Figure 2B, C*). Next, to isolate neuronal depolarization induced by *T. crassiceps* homogenate, but not mediated by K⁺, a cesium-based internal solution was used. Despite the addition of cesium the *T. crassiceps* homogenate still resulted in a large depolarization of the membrane potential (median 13.76, IQR 10.86–17.24 mV, *N* = 49 from 48 slices) which was larger than that caused by 11.4 mM K⁺ (p < 0.0001, Kruskal–Wallis test with Dunns' multiple comparison test, *Figure 2B, C*) but was not statistically different from that caused by *T. crassiceps* homogenate without cesium internal solution (p = 0.25, Kruskal–Wallis test with Dunns' multiple comparison test, *Figure 2B, C*). Together, this indicates that although K⁺ in the *T.*

*crassiceps* homogenate may contribute to membrane depolarization, much of the effect is mediated by a different component.

To determine the fraction of the *T. crassiceps* homogenate which underlies the acute excitatory effect on neurons we observed, we used a dialysis membrane to separate the fraction of the homogenate bigger than 3 kDa from the total homogenate. This removed molecules smaller than 3 kDa from the homogenate. When this dialysed *T. crassiceps* was puffed onto the cells the depolarizing response was greatly reduced. The median positive shift in membrane potential for dialysed *T. crassiceps* homogenate was only 0.27 mV (IQR 0.20–0.85 mV, *N* = 5 from 4 slices) as compared to a median for the total homogenate which was 13.76 mV (IQR 10.87–17.24 mV, *N* = 49 from 48 slices, p = 0.0002, Kruskal–Wallis test with Dunn's multiple comparison test, *Figures 1 and 2*). Phosphate-buffered saline (PBS), the solvent for the homogenate also did not produce substantial depolarization (median 1.00 mV, IQR 0.58–1.56 mV, *N* = 8 from 2 slices,) when compared to the dialysed *T. crassiceps* homogenate (p > 0.10, Kruskal–Wallis test with Dunn's multiple comparison test, *Figure 2D, E*). This suggests that the excitatory component of the *T. crassiceps* homogenate is a molecule smaller than 3 kDa.

Given that helminths are well known to excrete or secrete (E/S) products into their immediate environment, we next sought to determine whether the <3 kDa small molecule/s from *T. crassiceps* homogenate, which was found to induce neuronal depolarization, was also produced as an E/S product by *T. crassiceps* larvae. The media was collected over a period of 21 days and either used unaltered (*T. crassiceps* total E/S products) or buffer exchanged into a >3 kDa fraction using an Amicon stirred cell. Brief application of total *T. crassiceps* E/S products using a soma directed puff was sufficient to cause neuronal depolarization (median 12.40 mV, IQR 9.94–14.07 mV, *N* = 7 from 3 slices) significantly larger than that of the media control (median –0.61 mV, IQR –0.80–0.35 mV, *N* = 7 from 2 slices, p ≤ 0.0001, Kruskal–Wallis test with Dunn's multiple comparison test, *Figure 2F, G*, *Figure 2—source data 1*). However, the *T. crassiceps* E/S product fraction larger than 3 kDa (median 1.25 mV, IQR 1.02–2.21 mV, *N* = 8 from 1 slice) did not generate significant neuronal depolarization as compared to media control (p = 0.17, Kruskal–Wallis test with Dunn's multiple comparison test, *Figure 2F, G*). Together this set of experiments demonstrated that both *T. crassiceps* homogenate and *T. crassiceps* E/S products contain an excitatory component, which is a small molecule.

## The excitatory effects of *T. crassiceps* are mediated by glutamate receptor activation

*Robinson et al., 2012* have found Substance P, an abundant neuropeptide and neurotransmitter, in close vicinity to human NCC granulomas (*Robinson et al., 2012*). We therefore investigated whether Substance P could elicit a similar neuronal depolarizing response to that of *T. crassiceps* homogenate. We found, however, that 100 µM Substance P had no acute effect on the membrane potential of CA3 hippocampal pyramidal neurons (median 0.08 mV, IQR 0.01–0.12 mV, *N* = 5 from 1 slice, p = 0.0625, Wilcoxon signed rank test with theoretical median, *Figure 3A, B*, *Figure 3—source data 1*).

Next, we tested whether nicotinic acetylcholine receptors (nAchRs) - well described ionotropic receptors in the nervous system (*Bear et al., 2007*) - could be mediating neuronal depolarization. Blockade of nAchRs with mecamylamine hydrochloride (10 µM) did not, however, significantly alter the *T. crassiceps* homogenate induced depolarization. The median *T. crassiceps* homogenate induced depolarization was 16.28 mV (IQR 13.54–23.63 mV) during baseline, 16.58 mV (IQR 11.21–24.50 mV) in the presence of mecamylamine hydrochloride, and 13.20 mV (IQR = 10.37–23.48 mV) following washout (*N* = 5 from 5 slices, p = 0.6914, Friedman test, *Figure 3C, D*, *Figure 3—source data 1*). We also investigated the involvement of acid-sensing ion channels (ASICs) - proton-gated sodium channels known to be expressed by hippocampal neurons that induce neuronal depolarization when activated. Blockade of ASICs with the non-specific ASIC blocker amiloride (2 mM) did not, however, significantly attenuate the effect of the homogenate (*Figure 3E*), with the median *T. crassiceps* homogenate induced depolarization being 13.86 mV (IQR 11.00–16.48 mV) during baseline, 11.99 mV (IQR 4.97–19.03 mV) in the presence of amiloride, and 15.79 mV (IQR 13.25–24.67 mV) following washout (*N* = 10 from 10 slices, p = 0.1873, Friedman test, *Figure 3E, F*, *Figure 3—source data 1*).

Ionotropic glutamate receptors (GluRs) including α-amino-3-hydroxy-5-methyl-4-isoxazoleproprionic acid (AMPA), kainate, and *N*-methyl-D-aspartate (NMDA) receptors (*Bear et al., 2007*) are the major source of fast neuronal depolarization in neurons. It is plausible that *T. crassiceps* homogenate could

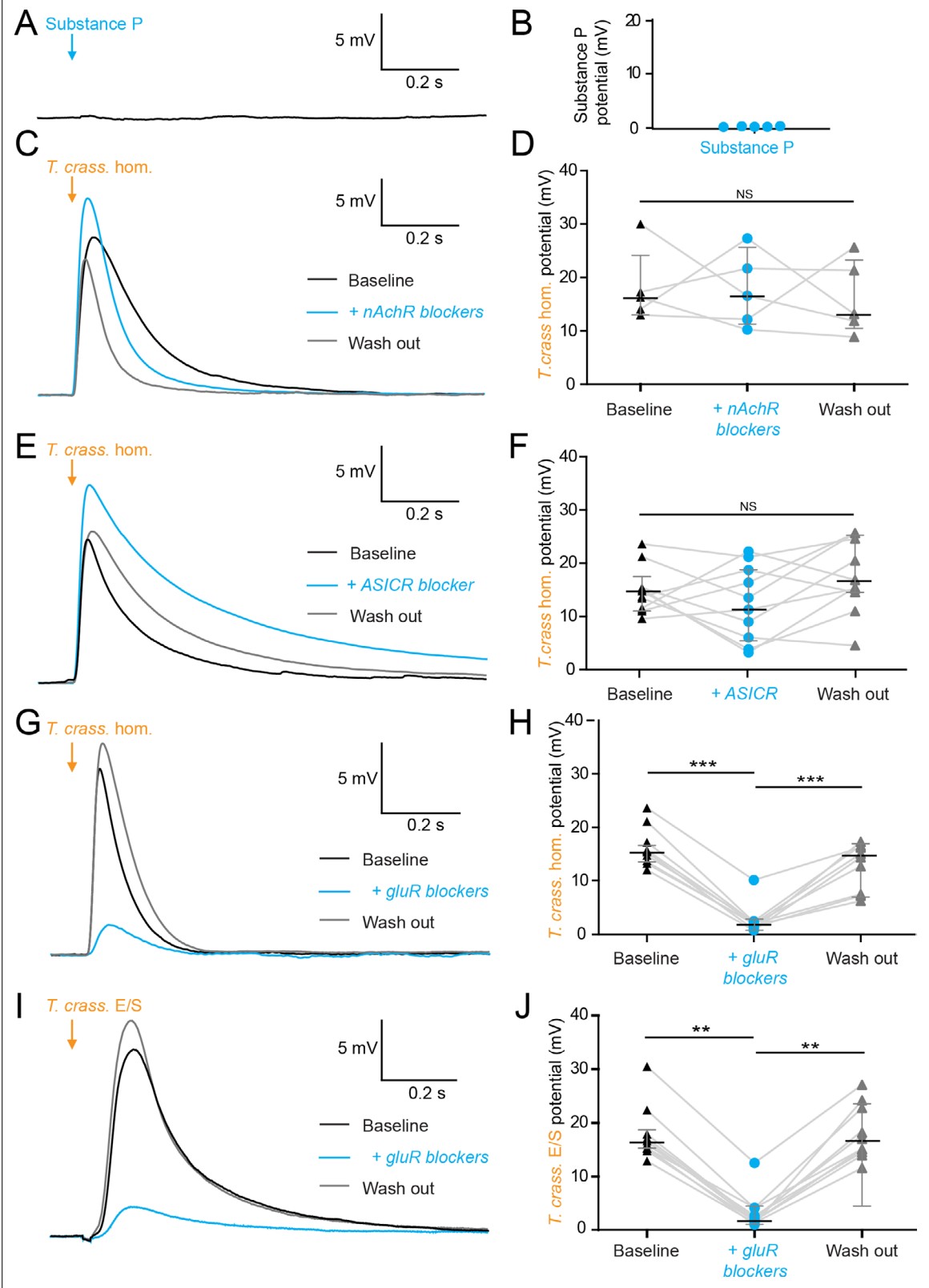

**Figure 3.** The excitatory effects of *Taenia crassiceps* are mediated by glutamate receptor activation. Whole-cell patch-clamp recordings in current-clamp were made from CA3 pyramidal neurons in rat organotypic hippocampal slices using a cesium-based internal in the presence of 2 µM tetrodotoxin (TTX). (**A**) 20-ms puff of Substance P (100 µM) via a puff pipette directed at the soma did not affect neuronal membrane potential. (**B**) Population data for Substance P application. (**C**) The depolarization in response to a *T. crassiceps* homogenate (*T. crass.* hom.) puff before (black trace), in the

*Figure 3 continued on next page*

**Figure 3 continued**

presence of (blue trace), and following washout (grey trace), of a nicotinic acetylcholine receptor (nAchR) blocker (mecamylamine hydrochloride, 10 µM). (**D**) Population data demonstrating that nAchR blockade has no significant effect on *T. crass*. hom. induced depolarization, NS = not significant. (**E**) The depolarization in response to a *T. crass* hom. puff before (black trace), in the presence of (blue trace), and following washout (grey trace), of an acid-sensing ion channel (ASIC) receptor blocker (amiloride, 2 mM). (**F**) Population data demonstrating that ASIC receptor blockade has no significant effect on *T. crass*. hom. induced depolarization, NS = not significant. (**G**) The depolarization in response to a *T. crass*. hom. puff before (black trace), in the presence of (blue trace), and following washout (grey trace), of a pharmacological cocktail to block glutamate receptors (10 µM CNQX, 50 µM D-AP5, and 2 mM kynurenic acid). (**H**) Population data showing that the depolarization response to *T. crass*. hom. is significantly reduced in the presence of glutamate receptor blockers and returns upon washout, ***p ≤ 0.001. (**I**) The depolarization in response to a puff of *T. crass* excretory/secretory (E/S) products is also markedly reduced during glutamate receptor blockade. (**J**) Population data showing that the depolarization response to *T. crass*. hom. is significantly reduced in the presence of glutamate receptor blockers and returns upon washout, **p ≤ 0.01.

The online version of this article includes the following source data for figure 3:

**Source data 1.** Patch clamp membrane voltages, cell properties, and metadata for all cells.

cause neuronal depolarization by activation of GluRs. To test this possibility, *T. crassiceps* homogenate induced neuronal depolarization was measured before, during, and after, the application of 10 µM CNQX, 50 µM D-AP5, and 2 mM kynurenic acid (in the aCSF) to block all three classes of GluRs (all in the presence of tetrodotoxin [TTX] and a cesium internal solution). We found that GluR blockade significantly reduced the median *T. crassiceps* homogenate induced neuronal depolarization from 14.72 mV (IQR 13.39–15.68 mV) to 1.90 mV (IQR 1.09–2.21 mV), which recovered to a median value of 14.37 mV (IQR 7.32–16.38 mV) following washout ($N = 9$ from 9 slices, $p \leq 0.01$, Friedman test with Dunn's multiple comparison test, *Figure 3G, H*, *Figure 3—source data 1*). Similarly, for *T. crassiceps* E/S products we found that application of GluR antagonists reduced the median total *T. crassiceps* E/S products-induced depolarization from 16.35 mV (IQR 15.17–19.03 mV) to 2.19 mV (IQR 1.67–4.16 mV), which recovered to a median value of 16.43 mV (IQR 14.41–23.14 mV) following washout ($N = 10$ from 10 slices, $p \leq 0.01$, Friedman test with Dunn's multiple comparison test, *Figure 3I, J*, *Figure 3—source data 1*). These results indicate that the acute excitatory effects of both *T. crassiceps* homogenate and *T. crassiceps* E/S products are mediated through GluRs.

## *T. crassiceps* products contain excitatory amino acids that are detected by iGluSnFR

Glutamate is the prototypical agonist of GluRs (*Bear et al., 2007*). We therefore sought to directly detect glutamate in *T. crassiceps* larval products using glutamate-sensing fluorescent reporters. We utilized the genetically encoded glutamate reporter iGluSnFR, which was virally transfected into mouse organotypic hippocampal slice cultures under the synapsin 1 gene promoter and imaged using wide-field epifluorescence microscopy (*Figure 4A* and Materials and methods). Brief puffs of *T. crassiceps* homogenate delivered over the soma of iGluSnFR-expressing pyramidal neurons caused a robust increase in fluorescence (median 23.32%, IQR 5.79% to 37.66%, $N = 35$ from 6 slices, *Figure 4B, G*, *Figure 4—source data 1*) when compared to puffs of aCSF (median 0.0012%, IQR –0.42% to 1.02%, $N = 12$ from 1 slice, $p < 0.0001$, Kruskal–Wallis test with Dunn's multiple comparison test, *Figure 4G*). Fluorescence increases were also observed with puffs of *T. crassiceps* E/S products (median 11.69%, IQR 2.94% to 15.85%, $N = 10$ from 2 slices, *Figure 4C, G*) although this did not reach statistical significance when compared to puffs of aCSF (median 0.0012%, IQR –0.42% to 1.02%, $N = 12$ from 1 slice, $p = 0.16$, Kruskal–Wallis test with Dunn's multiple comparison test, *Figure 4G*). Aspartate has a similar chemical structure to glutamate and is also known to excite neurons via glutamate receptor activation (*Dingledine and McBain, 1999*). iGluSnFR is also known to be sensitive to aspartate as well as glutamate (*Marvin et al., 2013*). We found that puffing aCSF containing 100 µM glutamate (median 31.35%, IQR 15.35% to 54.92%, $N = 14$ from 2 slices, *Figure 4E, G*) or 100 µM aspartate (median 15.09%, IQR 11.12% to 33.12%, $N = 5$ from 1 slice, *Figure 4F, G*) onto hippocampal pyramidal neurons expressing iGluSnFR resulted in a significantly higher fluorescence as compared to aCSF (median 0.0012%, IQR –0.42% to 1.02%, $N = 12$ from 1 slice, Kruskal–Wallis test with Dunn's multiple comparison test, $p \leq 0.05$, *Figure 4G*). Similarly, puffing aCSF containing 100 µM glutamate (median 15.05 mV, IQR 11.12–18.71 mV, $N = 6$ from 1 slice, *Figure 4H, I*, *Figure 4—source data 2*) or 80 µM aspartate (median 7.75 mV, IQR 3.74–15.99 mV, $N = 5$ from 1 slice, *Figure 4H, I*) during whole-cell current-clamp recordings elicited membrane depolarizations which were similar those observed with

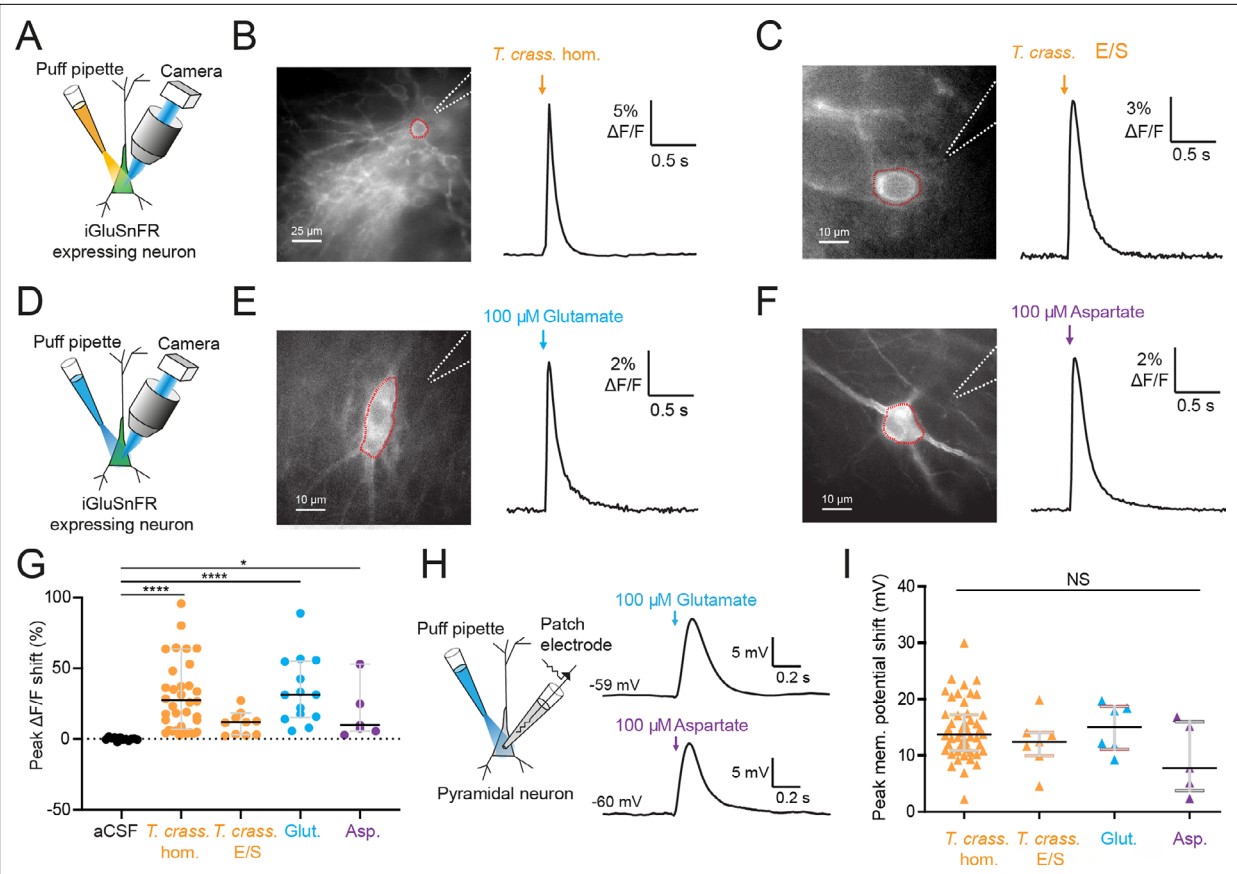

**Figure 4.** *Taenia crassiceps* products contain excitatory amino acids detected using iGluSnFR. (**A**) Schematic of the experimental setup for glutamate detection by the genetically encoded fluorescent glutamate reporter iGluSnFR, including puff pipette and sCMOS camera for imaging following excitation using a 475/28 nm LED-based light engine. (**B**) Left, iGluSnFR-expressing neuron with the region of interest (red dashed line) used to calculate d*F/F* trace (right) during a 20-ms *T. crassiceps* homogenate (*T. crass.* hom.) puff (orange arrow). (**C**) As in 'B' but with a 20-ms *T. crassiceps* excretory/secretory (*T. crass.* E/S) product puff (orange arrow). (**D**) Schematic of experimental setup as in 'A' but with glutamate or aspartate application via puff pipette. (**E**) iGluSnFR fluorescence just after a 20-ms puff of artificial cerebrospinal fluid (aCSF) containing 100 μM glutamate (blue arrow). (**F**) As in 'E' but with iGluSnFR fluorescence during a 20-ms puff of aCSF containing 100 μM aspartate (purple arrow). (**G**) Population data comparing peak d*F/F* shifts after a 20-ms puff of aCSF (as a negative control), *T. crass.* hom., *T. crass.* E/S products, aCSF containing 100 μM glutamate (as a positive control), or aCSF containing 100 μM aspartate (as a positive control), values with medians ± IQR, *p ≤ 0.05, ****p ≤ 0.0001. (**H**) Schematic of whole-cell patch-clamp recording in current-clamp mode from a CA3 pyramidal neuron using a cesium-based internal and in the presence of 2 μM tetrodotoxin (TTX). 20-ms puff of aCSF containing 100 μM glutamate produces a significant depolarizing shift in membrane potential (top trace), a similar puff but with 100 μM aspartate elicited a similar response (bottom trace). (**I**) Population data comparing peak membrane potential shift after a 20-ms puff of *T. crass.* hom., *T. crass.* E/S, aCSF containing 100 μM glutamate, and aCSF containing 100 μM aspartate. Values with medians ± IQR; NS = not significant.

The online version of this article includes the following source data for figure 4:

**Source data 1.** iGluSnfr fluorescence values and metadata for all cells.

**Source data 2.** Patch clamp membrane voltages, cell properties, and metadata for all cells.

*T. crassiceps* homogenate (median 13.76 mV, IQR 10.87–17.24 mV, *N* = 49 from 48 slices, *Figure 4I*) and E/S products (median 12.40 mV, IQR 9.94–14.07 mV, *N* = 7 from 3 slices, Kruskal–Wallis test with Dunn's multiple comparison test, p > 0.05, *Figure 4I*).

## *T. crassiceps* larvae contain and produce glutamate and aspartate

Given the short-term responses to homogenate and E/S products we observed using iGluSnFR, we next sought to record possible evidence of changes in ambient glutamate or aspartate using a more naturalistic setting where iGluSnFR-expressing neurons were placed adjacent to a live *T. crassiceps* larva (see Materials and Methods, *Figure 5A–C*). During imaging periods lasting up to 15 min, fluorescence was stable, indicating no detectable oscillatory changes in ambient glutamate/aspartate on the

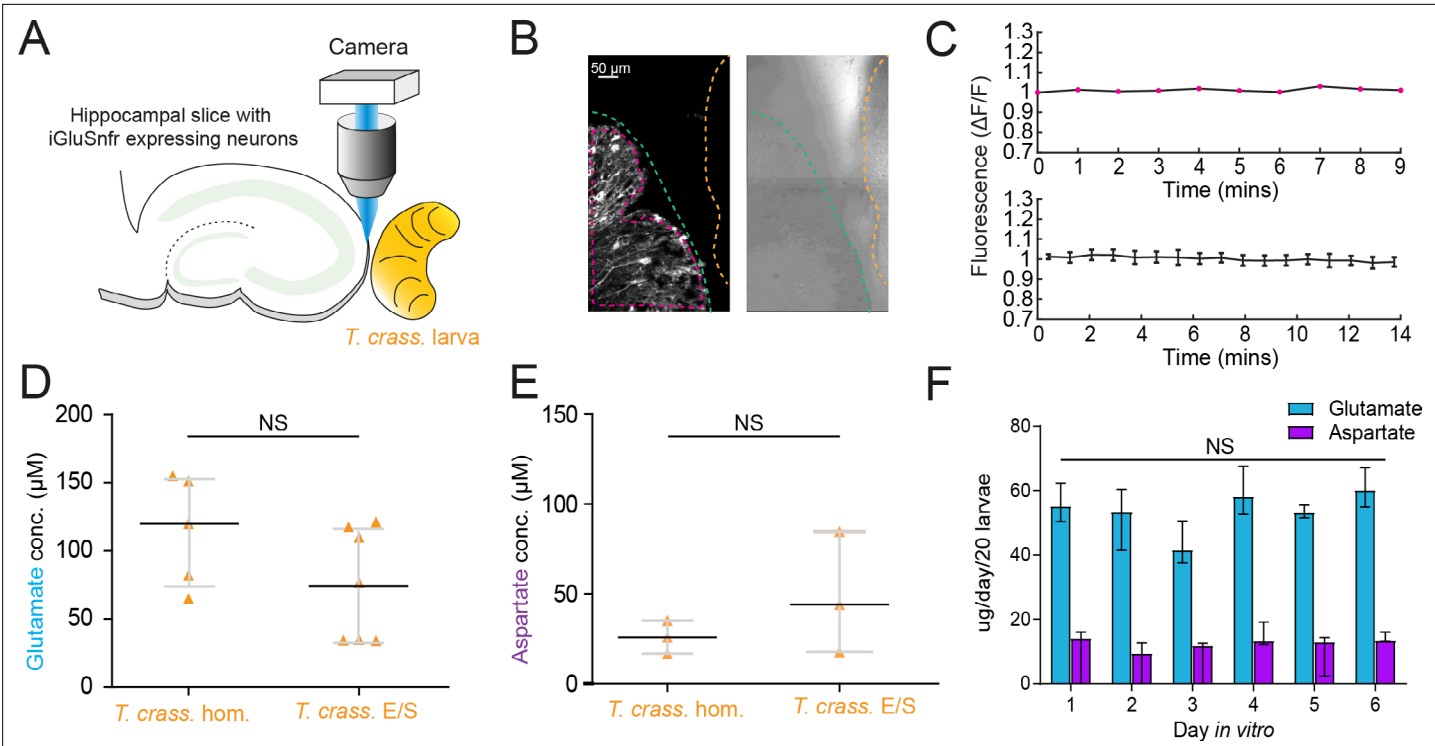

**Figure 5.** *Taenia crassiceps* larvae contain and produce glutamate and aspartate. (**A**) Schematic of the experimental setup whereby a living *Taenia crassiceps* (*T. crass.*) larva is placed in close proximity to a hippocampal organotypic brain slice with iGluSnFR-expressing neurons, for detection of glutamate fluctuations. (**B**) Left, fluorescence image of a slice with iGluSnFR-expressing neurons (green dashed line) adjacent to a *T. crass.* larva (orange dashed line), also visible in the transmitted light image (right). The region of interest (pink dashed line) was used to calculate the top dF/F trace in 'C'. (**C**) Top, dF/F trace from the example in 'B' during a 9-min recording. Lower trace: population data of the mean ± SEM from five slice-larva pairs showed no detectable oscillations or changes in glutamate. (**D**) Glutamate concentration in *T. crass.* Homogenate (*T. crass.* hom.) and *T. crass.* excretory/secretory (*T. crass.* E/S) products as measured using a glutamate assay kit, values with medians ± IQR; NS = not significant. (**E**) Aspartate concentration in *T. crass.* hom. and *T. crass.* E/S as measured using an aspartate assay kit, values with medians ± IQR; NS = not significant. (**F**) Population data showing *de novo* production of glutamate and aspartate by by by *T. crass* larvae for 6 days *in vitro*, values with medians ± IQR; NS = not significant.

The online version of this article includes the following source data for figure 5:

**Source data 1.** Glutamate and aspartate values and metadata for all samples.

timescales recorded (*N* = 2 slices, *Figure 5B, C*). Our submerged recording setup might have led to swift diffusion or washout of released glutamate, possibly explaining the lack of observable changes. We then directly measured the concentration of glutamate and aspartate in our *T. crassiceps* products using separate glutamate and aspartate assays (see Materials and methods). We found that the median concentration of glutamate in the *T. crassiceps* homogenate and E/S products were 119.9 µM (IQR 73.41–153.2 µM, *N* = 5, *Figure 5D*, *Figure 5—source data 1*) and 72.32 µM (IQR 32.95–116.0 µM, *N* = 6, p = 0.1061, Mann–Whitney test, *Figure 5D*), respectively. The median concentration of aspartate in *T. crassiceps* homogenate and E/S products were 25.8 µM (IQR 16.5–35.1 µM, *N* = 3, *Figure 5E*) and 44.03 µM (IQR 17.56–84.74 µM, *N* = 3, p = 0.4000, Mann–Whitney test, *Figure 5E*), respectively. Together, these findings suggest that *T. crassiceps* larval homogenate and E/S products contain the excitatory amino acids glutamate and aspartate at concentrations that are sufficient to elicit neuronal depolarization.

Next, to determine whether *T. crassiceps* larvae actively produce and excrete/secrete glutamate and/or aspartate into their environments, we measured the *de novo* daily production of glutamate and aspartate by larvae following harvest of live larvae. *T. crassiceps* larvae released a relatively constant median daily amount of glutamate, ranging from 41.59 to 60.15 µg/20 larvae, which showed no statistically significant difference across days 1–6 (*N* = 3 per day, p = 0.18, Kruskal–Wallis test, *Figure 5F*). Similarly, *T. crassiceps* larvae released a relatively constant median daily amount of aspartate, ranging

from 9.43 to 14.18 µg/20 larvae, which showed no statistically significant difference across days 1–6 ($N$ = 3 per day, p = 0.28, Kruskal–Wallis test, *Figure 5F*).

### *T. solium* larvae depolarize human neurons via the production of glutamate

*T. crassiceps*, whilst a closely related species to *T. solium*, is not the causative pathogen in humans. We therefore sought to determine whether homogenate and E/S products from *T. solium* (the natural pathogen in humans), also have an excitatory effect on human neurons in human brain tissue (the natural host). During whole-cell patch-clamp recordings made from human layer 2/3 pyramidal cells in acute cortical brain slices, pico-litre volumes of *T. solium* larval homogenate and *T. solium* larval E/S products were delivered by a glass pipette. Similarly to *T. crassiceps*, both *T. solium* homogenate (*Figure 6A*) and *T. solium* E/S products (*Figure 6A*) caused membrane depolarization, which was sufficient to result in action potential firing.

We then measured the glutamate and aspartate concentrations in *T. solium* homogenate and *T. solium* E/S products. Significant levels of glutamate were detected in both *T. solium* homogenate (46.4 µM, $N$ = 1, *Figure 6C*, *Figure 6—source data 1*) and the *T. solium* E/S products (722 µM, $N$ = 1, *Figure 6C*). Aspartate levels were, however, undetectable in the *T. solium* homogenate ($N$ = 1, *Figure 6C*), and low in the E/S products (15.3 µM, $N$ = 1, *Figure 6C*). Finally, we sought to determine whether *T. solium* larvae, like *T. crassiceps* larvae, actively produce and excrete/secrete glutamate and/or aspartate. We found that, following harvest of live larvae, *T. solium* larvae released a large amount of glutamate on the first day post-harvest (391.1 µM/20 larvae, $N$ = 1, *Figure 6D*), none on the second and third day *in vitro* ($N$ = 1 each, *Figure 6D*), but produced significant amounts of glutamate *de novo* on days 4–6 (day 4: 131.9 µM/20 larvae, $N$ = 1; day 5: 140.6 µM/20 larvae, $N$ = 1; day 6: 158.8 µM/20 larvae, $N$ = 1, *Figure 6D*). Aspartate was not detected on days 1 and 3 ($N$ = 1 each, *Figure 6D*), but was observed to be released in smaller amounts than glutamate on days 2 and 4–6 (day 2: 8.4 µM/20 larvae, $N$ = 1; day 4: 16.8 µM/20 larvae, $N$ = 1; day 5: 28.1 µM/20 larvae, $N$ = 1; day 6: 31.9 µM/20 larvae, $N$ = 1, *Figure 6D*). These results demonstrate that *T. solium* larvae continually release glutamate and aspartate into their immediate surroundings.

## Discussion

Here, we used patch-clamp electrophysiology, and calcium imaging in rodent hippocampal organotypic slice cultures and human acute cortical slices to demonstrate that cestode larval products cause neuronal depolarization and can initiate SLEs via glutamate receptor activation. Glutamate-sensing fluorescent reporters and amino acid assays revealed that *T. crassiceps* and *T. solium* larvae contain and release the excitatory amino acid glutamate and, to a lesser extent, aspartate.

Clinical evidence and animal models conclusively demonstrate that the presence of cestode larvae in the brain can result in the generation of seizures (*Nash et al., 2015*; *Verastegui et al., 2015*). Previous work has focused on the involvement of the host inflammatory response in seizure generation (*Robinson et al., 2012*; *Stringer et al., 2003*). Whilst this is likely important, it does not preclude the involvement of additional or exacerbating pathogenic mechanisms for seizure generation and epileptogenesis in NCC. In this study we have identified that *Taenia* larvae have a direct excitatory effect on neurons via glutamatergic signalling. This is important, as the central role of glutamatergic signalling in epileptogenesis has conclusively been shown using multiple cell culture (*Sun et al., 2001*; *Sombati and Delorenzo, 1995*; *DeLorenzo, 1998*), slice (*Anderson et al., 1986*; *Stasheff et al., 1989*; *Ziobro et al., 2011*), and *in vivo* models of epilepsy (*Croucher and Bradford, 1990*; *Croucher et al., 1988*; *Rice and DeLorenzo, 1998*).

Clinically, beyond NCC, other major causes of adult-acquired epilepsy are stroke, traumatic brain injury and CNS tumours (*Hauser et al., 1940*; *Forsgren et al., 2005*). In these other causes of acquired epilepsy, glutamatergic signalling, and glutamate excitotoxicity, are thought to be central to the pathogenic process. Glutamate excitotoxicity occurs when depolarized, damaged, or dying neurons release glutamate, which activate surrounding neurons via NMDA and AMPA receptors, resulting in sustained neuronal depolarization, $Ca^{2+}$ influx and the subsequent activation of signalling cascades and enzymes. These, in turn, lead to cell death via necrosis and apoptosis (*Ankarcrona et al., 1995*), the further release of intracellular glutamate, and the propagation of the excitotoxic

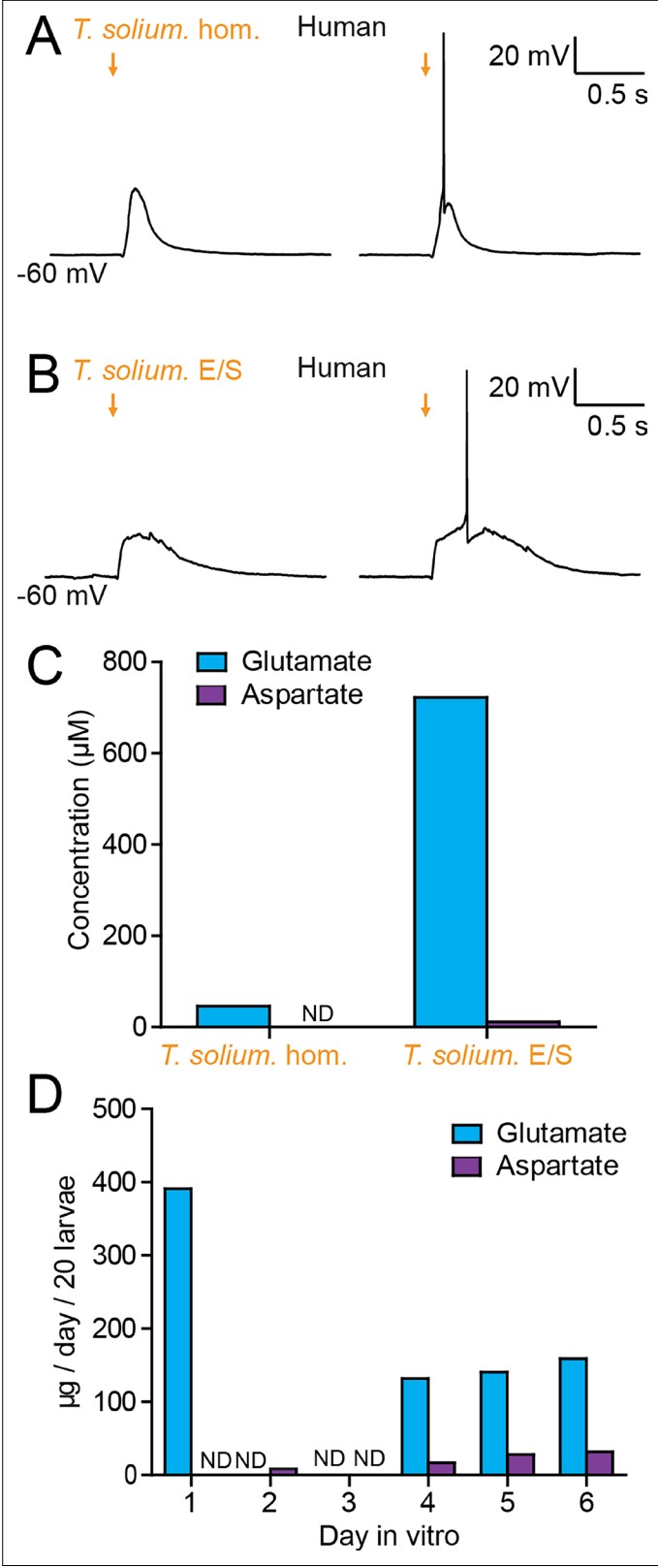

**Figure 6.** *Taenia solium* larvae depolarize human neurons via the production of glutamate. (**A**) Whole-cell patch-clamp recordings in current-clamp from a layer 2/3 human frontal lobe cortical pyramidal neuron in an acute brain slice, whilst increasing amounts of *T. solium* homogenate (*T. solium* hom.) was applied via a puff pipette (left to right, orange arrows). Small amounts of *T. solium* hom. resulted in depolarization (left), whilst an increased amount

*Figure 6 continued on next page*

*Figure 6 continued*

elicited an action potential (right). (**B**) As in 'A', *T. solium* excretory/secretory (*T. solium* E/S) products elicited membrane depolarization and an increased amount elicited an action potential. (**C**) Concentrations of glutamate and aspartate in *T. solium* hom. and *T. solium* E/S ($N = 1$, ND = not detectable). (**D**) *De novo* production of glutamate and aspartate by *T. solium* larvae over a 6-day culture period ($N = 1$, ND = not detectable).

The online version of this article includes the following source data for figure 6:

**Source data 1.** Patch clamp cell properties and metadata for A&B and glutamate and aspartate values and metadata for all samples in C&D.

process to neighbouring cells. In the neurons that survive, the prolonged exposure to glutamate has been shown to cause hyperexcitability and seizures (*Sun et al., 2001*; *DeLorenzo, 1998*) via multiple mechanisms, including enhanced intrinsic excitability and NMDA receptor-dependent disruption of GABAergic inhibition (*Lee et al., 2011*; *Terunuma et al., 2010*; *Buckingham et al., 2011*). Given our finding that cestode larvae contain and release significant quantities of glutamate, it is possible that the brain's homeostatic mechanisms for taking up and metabolizing excess glutamate are overwhelmed by larval-derived glutamate released into the extracellular space. Therefore, similar glutamate-dependent excitotoxic and epileptogenic processes that occur in stroke, traumatic brain injury, and CNS tumours are likely to also occur in NCC.

Glioma, a common adult primary brain tumour, which typically presents with seizures in over 80% of patients (*Santosh and Sravya, 2017*), is an intriguing condition for comparison to NCC. Here, the tumour cells themselves have been shown to release glutamate into the extracellular space via the system $x_c^-$ cystine–glutamate antiporter (*Buckingham et al., 2011*). Interestingly, there is compelling evidence that tumoural release of glutamate via this mechanism both causes seizures and favours glioma preservation, progression, and invasion in cases of malignant glioma (*Sontheimer, 2008*; *Takano et al., 2001*; *Huang et al., 2013*). Analogously, it is possible that *Taenia* larvae in the brain utilize the release of glutamate and the induction of glutamate excitotoxicity to facilitate their growth and expansion, with the accompanying effect of seizure generation. In addition, the death of *Taenia* larvae, or larval-derived cells, would also result in the release of metabolic glutamate, further contributing to glutamate release and excitotoxicity. In the case of glioma, where the mechanism of glutamate release by tumour cells is known, pharmacological agents (e.g. sulfasalazine), which block glutamate release have considerable potential as therapeutic agents for reducing seizure burden in this condition. Therefore, it is important that future work attempts to identify the molecular mechanisms underlying the *Taenia*-specific production of glutamate to inform the development of new therapeutic strategies to potentially reduce larval expansion and possibly treat seizures in NCC.

An important consideration is how we might reconcile our findings of glutamate release by *Taenia* larvae with the clinical picture of delayed symptom onset in NCC. In people with NCC, some experience acute seizures immediately following infection whilst others present with seizures months to years following initial infection (*Adalid-Peralta et al., 2013*; *Verma et al., 2011*). This suggests that multiple different, or possibly interacting, mechanisms might be involved in the epileptogenic process in NCC. A recent report has shown that products from *T. solium* cysts mimicking viable cysts contain the enzyme glutamate dehydrogenase, which metabolizes glutamate, whilst products mimicking degenerating cysts do not (*Prodjinotho et al., 2022*). This predicts that the glutamate levels the brain would be exposed to would be highest when a cyst dies and releases its contents into its immediate surroundings. Neuroinflammation has long been thought to be central to the development of seizures in NCC. In support of this, larval suppression of host inflammatory responses could help explain why many patients may remain asymptomatic for years following infection. Our findings are not at odds with this line of thinking when considering the effect of neuroinflammation on extracellular glutamate uptake. In the healthy brain parenchymal astrocytes are optimized to maintain extracellular glutamate at low concentrations (*Perea and Araque, 2010*). However, the inflammatory transition of astrocytes to a reactive phenotype is known to impede their ability to buffer uptake of extracellular glutamate (*Seifert et al., 2010*). It is therefore conceivable that cyst-associated reactive astrocytosis may gradually erode the ability of homeostatic mechanisms to compensate for larvae derived extracellular glutamate resulting in delayed symptom onset in NCC.

There is still considerable uncertainty regarding the precise sequence of events leading to seizure onset in NCC. Nonetheless our findings provide the first evidence that, as is the case with the other

common causes of adult-acquired epilepsy (i.e. stoke, traumatic brain injury, and glioma), increased extracellular, parasite-derived excitatory amino acids such as glutamate, and perturbed glutamatergic signalling, possibly play a role in the development of seizures in NCC.

## Materials and methods

### *Taenia* maintenance and preparation of whole cyst homogenates and E/S products

Larvae of *T. crassiceps* (ORF strain) were donated to us by Dr Siddhartha Mahanty (University of Melbourne, Melbourne, Australia) and propagated *in vivo* by serial intraperitoneal infection of 5- to 8-week-old female C57BL/6 mice. Every 3 months parasites were harvested by peritoneal lavage and washed once in PBS (1×, pH) containing penicillin (500 U/ml); streptomycin (500 µg/ml); and genta-micin sulphate (1000 ug/ml) (Sigma-Aldrich), before being washed a further five times in standard PBS (1×).

For the preparation of *T. crassiceps* whole cyst homogenate, larvae were stored immediately after harvesting at –80°C. Later the larvae were thawed (thereby lysing the cells) and suspended in PBS (1×) containing a protease cocktail inhibitor (1% vol/vol, Sigma-Aldrich) at a larval:PBS ratio of 1:3. The larvae were then homogenized on ice using a Polytron homogenizer (Kinematica). The resulting mixture was centrifuged at 4000 rpm for 20 min at 4°C. The liquid supernatant (between the white floating layer and solid pellet) was collected and sterile filtered through a 0.22-µm size filter (Millex-GV syringe filter, Merck). This supernatant was then aliquoted and stored at –80°C until use. To assess whether large or small molecules were responsible for the excitation of neurons, a portion of the whole cyst homogenate was dialysed using a Slide-A-Lyzer dialysis cassette (3-kDa MWCO, Separa-tions) in 2 l of aCSF at 4°C. The aCSF solution was changed twice over 24 hr. To determine the ionic composition of *T. crassiceps* whole cyst homogenate a Cobas 6000 analyser (Roche) was used, with ion-specific electrodes for $K^+$ and $Na^+$. A Mettler Toledo SevenCompact pH meter S210 (Merck) was used to determine the pH of the homogenate.

For the preparation of *T. crassiceps* excretory/secretory products, after harvesting, washed larvae were maintained for 14–21 days at 37°C in 5% carbon dioxide ($CO_2$) in 50 ml tissue culture flasks (approximately 7 ml volume of larvae per flask) containing culture medium consisting of Earle's balanced salt solution (EBSS) supplemented with glucose (5.3 g/l), Glutamax (1×), penicillin (50 U/ml), streptomycin (50 µg/ml), gentamicin sulphate (100 µg/ml), and nystatin (11.4 U/ml) (Sigma-Aldrich). Culture medium was replaced after 24 hr, and the original media discarded. Thereafter, culture media were replaced and collected every 48–72 hr, stored at –20°C and at the end of the culture period, thawed and pooled. The pooled conditioned medium was termed total excretory/secretory products (total E/S). A portion of the total E/S was aliquoted and stored at –80°C, whilst another portion was concentrated (about 100×) and buffer exchanged to PBS using an Amicon stirred cell with a 3-kDa MWCO cellulose membrane (Sigma-Aldrich). The concentrated product contained *T. crassiceps* E/S products larger in size than 3 kDa (E/S >3 kDa), in PBS (1×). This was aliquoted and stored at –80°C. The fraction of the total E/S that passed through the 3-kDa membrane (E/S <3 kDa, still in medium) was also retained, aliquoted and stored at –80°C.

For the preparation of *T. solium* whole cyst homogenate, larvae of *T. solium* were harvested from the muscles of a heavily infected, freshly slaughtered pig. After extensive washing with sterile 1× PBS, *T. solium* larvae were suspended in PBS containing phenylmethyl-sulphonyl fluoride (5 mM) and leupeptin (2.5 µM) at a larvae:PBS ratio of 1:3. Larvae were then homogenized using a sterile handheld homogenizer at 4°C. The resulting homogenate was sonicated (4 × 60 s at 20 kHz, 1 mA, with 30-s intervals) and gently stirred with a magnetic stirrer for 2 hr at 4°C. Thereafter it was centrifuged at 15,000 × *g* for 60 min at 4°C and the liquid supernatant (between the white floating layer and solid pellet) was collected. The supernatant was filtered through 0.45-µm size filters (Millex-GV syringe filter, Merck), aliquoted and stored at −80°C. All *T. crassiceps* and *T. solium* larval products were assessed for protein concentration using a BCA protein or Bradford's assay kit, respectively.

For the assessment of daily glutamate and aspartate production both *T. crassiceps* and *T. solium* larvae were placed into 6-well plates (±20 per well, of roughly 5 mm length) with 2 ml of culture medium (*T. solium* medium: RPMI 1640 with 10 mM HEPES buffer, 100 U/ml penicillin, 100 µg/ml streptomycin, 0.25 µg/ml amphotericin B, and 2 mM L-glutamine; *T. crassiceps* medium: EBSS with

100 U/ml penicillin, 100 µg/ml streptomycin, 11.4 U/ml nystatin, and 2 mM Glutamax). Every 24 hr, 1 ml of culture medium was collected from each well, stored at −80°C, and replaced with fresh culture medium. The concentrations of glutamate and aspartate were measured using a Glutamate assay kit and an Aspartate assay kit, respectively, according to the supplier's instructions (Sigma-Aldrich). '*T. solium* E/S' used in human organotypic brain slice electrophysiology experiments, consisted of a mixture of equal volumes of the *T. solium* medium collected on D1, D2, and D3.

## Brain slice preparation

Rodent hippocampal organotypic brain slices were prepared using 6- to 8-day-old Wistar rats and C57BL/6 mice following the protocol originally described by *Stoppini et al., 1991*. Brains were extracted and swiftly placed in cooled (4°C) dissection media consisting of EBSS (Sigma-Aldrich); 6.1 g/l HEPES (Sigma-Aldrich); 6.6 g/l D-glucose (Sigma-Aldrich); and 5 µM sodium hydroxide (Sigma-Aldrich). The hemispheres were separated, and individual hippocampi were removed and immediately cut into 350 µm slices using a McIlwain tissue chopper (Mickle). Cold dissection media was used to rinse the slices before placing them onto Millicell-CM membranes (Sigma-Aldrich). Slices were maintained in culture medium consisting of 25% (vol/vol) EBSS (Sigma-Aldrich); 49% (vol/vol) minimum essential medium (Sigma-Aldrich); 25% (vol/vol) heat-inactivated horse serum (Sigma-Aldrich); 1% (vol/vol) B27 (Invitrogen, Life Technologies); and 6.2 g/l D-glucose. Slices were incubated in a 5% $CO_2$ humidified incubator at between 35 and 37°C. Recordings were made after 6–14 days in culture.

For human cortical acute brain slices, brain tissue from temporal cortex was obtained from patients undergoing elective neurosurgical procedures at Mediclinic Constantiaberg Hospital to treat refractory epilepsy (two patients, details below).

| Age at surgery | Gender | Reason for surgery | Region | Tissue type | Epilepsy duration |
|---|---|---|---|---|---|
| 41 | Female | Refractory epilepsy | Temporal cortex | Access | 2 years |
| 4 | Male | Refractory epilepsy, malformation of cortical development | Temporal cortex | Least abnormal | Unknown |

Informed consent to use tissue for research purposes was obtained from patients or guardians prior to surgery. Resected brain tissue was transported from surgery to the laboratory in an ice-cold choline-based cutting solution composed of choline chloride (110 mM, Sigma-Aldrich); $NaHCO_3$ (26 mM, Sigma-Aldrich); D-glucose (10 mM, Sigma-Aldrich); sodium ascorbate (11.6 mM, Sigma-Aldrich); $MgCl_2$ (7 mM, Sigma-Aldrich); sodium pyruvate (3.1 mM, Sigma-Aldrich); KCl (2.5 mM, Sigma-Aldrich); $NaH_2PO_4$ (1.25 mM, Sigma-Aldrich); and $CaCl_2$ (0.5 mM, Sigma-Aldrich). Cutting solution was bubbled with carbogen gas (95% $O_2$, 5% $CO_2$) and had an osmolality of approximately 300 mOsm. In the lab, the pia mater was carefully removed, and the tissue was trimmed into blocks for sectioning using a Compresstome (Model VF-210-0Z, Precisionary Instruments). Slices were cut at a thickness of 350 µm and allowed to recover for ~30 min at 34°C in standard aCSF composed of NaCl (120 mM, Sigma-Aldrich); KCl (3 mM, Sigma-Aldrich); $MgCl_2$ (2 mM, Sigma-Aldrich); $CaCl_2$ (2 mM, Sigma-Aldrich); $NaH_2PO_4$ (1.2 mM, Sigma-Aldrich); $NaHCO_3$ (23 mM, Sigma-Aldrich); and D-glucose (11 mM, Sigma-Aldrich). The aCSF was adjusted to pH 7.4 using 0.1 mM NaOH (Sigma-Aldrich), was continuously carbogenated, and had an osmolality of ~300 mOsm. After recovery, slices were left for at least 1 hr at room temperature before recording.

## Electrophysiology, calcium and glutamate imaging

For experiments using calcium and glutamate imaging, mouse hippocampal organotypic brain slices were used. For all other experiments rat hippocampal organotypic brain slices were used. A subset of experiments used acute human cortical brain slices and are specified. Brain slices were transferred to a submerged recording chamber on a patch-clamp rig, which was maintained at a temperature between 28 and 30°C and were continuously superfused with standard aCSF bubbled with carbogen gas (95% $O_2$:5% $CO_2$, Afrox) using peristaltic pumps (Watson-Marlow). The standard aCSF was composed of NaCl (120 mM, Sigma-Aldrich); KCl (3 mM, Sigma-Aldrich); $MgCl_2$ (2 mM, Sigma-Aldrich); $CaCl_2$ (2 mM, Sigma-Aldrich); $NaH_2PO_4$ (1.2 mM, Sigma-Aldrich); $NaHCO_3$ (23 mM, Sigma-Aldrich); and D-glucose (11 mM, Sigma-Aldrich) in deionized water with pH adjusted to between 7.35 and 7.40 using 0.1 mM NaOH (Sigma-Aldrich). Neurons in the CA3 region of the hippocampus were visualized using a Zeiss Axioskop or Olympus BX51WI upright microscope using 20× or 40× water-immersion objectives and targeted for recording. Micropipettes were prepared (tip resistance between 3 and 7 MΩ) from

borosilicate glass capillaries (outer diameter 1.2 mm, inner diameter 0.69 mm, Harvard Apparatus Ltd) using a horizontal puller (Sutter). Recordings were made in current-clamp mode using Axopatch 200B amplifiers (Axon Instruments) and data acquired using WinWCP (University of Strathclyde) or Igor (Markram Laboratory, Ecole polytechnique fédérale de Lausanne). Matlab (MathWorks) was utilized for trace analysis. Basic properties of each cell were recorded (see Figure supplements). Cells were excluded from analyses if the Ra was greater than 80 Ω or if the resting membrane potential was above –40 mV. Two internal solutions were used: a 'standard' internal solution (K-gluconate (126 mM), KCl (4 mM), HEPES (10 mM), $Na_2ATP$ (4 mM), NaGTP (0.3 mM), and $Na_2$-phosphocreatine (10 mM); Sigma-Aldrich) and a 'cesium' internal solution (CsOH (120 mM), gluconic acid (120 mM), HEPES (40 mM), $Na_2ATP$ (2 mM), NaGTP (0.3 mM), and NaCl (10 mM); Sigma-Aldrich). Experimental substances were puffed onto neurons using an OpenSpritzer, a custom-made pressure ejection system (*Forman et al., 2016*). Current was injected if required to ensure a neuronal resting membrane potential within 2 mV of –60 mV. In all puffing experiments each data point represents the mean peak puff-induced change in membrane potential from 10 sweeps. For wash-in recordings drugs were washed in for 8 min before recordings were made. Drugs were washed out for at least 8 min before washout recordings were made. In some experiments, TTX (2 µM, Sigma-Aldrich) was added to the aCSF to block voltage-gated sodium channels.

For calcium and glutamate imaging, organotypic hippocampal mouse brain slices were virally transfected with a genetically encoded $Ca^{2+}$ reporter (GCAMP6s under the synapsin 1 gene promoter, AAV1.Syn.GCaMP6s.WPRE.SV40, Penn Vector Core) or a genetically encoded glutamate reporter (iGluSnFR under the human synapsin 1 gene promoter, pAAV.hSynapsin.SF-iGluSnFR.A184S, Addgene) 1 day post culture using the OpenSpritzer and imaged 5 days later using an Olympus BX51WI microscope, 20× water-immersion objective, CCD camera (CCE-B013-U, Mightex for the calcium imaging and an Andor Zyla 4.2 for the glutamate imaging). For the calcium imaging excitation was provided using a 470-nm LED (Thorlabs). For glutamate imaging, excitation was generated using an LED-based light engine, 475/28 nm (Lumencor). For both imaging paradigms a Chroma 39002 EGFP filter set (480/30 nm excitation filter, 505 nm dichroic, 535/40 nm emission filter) was utilized. Images were collected using µManager (*Edelstein et al., 2010*). Calcium imaging data were analysed using Caltracer3 beta. Glutamate imaging data were analysed using custom scripts written in Matlab.

For real-time imaging of glutamate signalling in neurons adjacent to a live *T. crassiceps* larva, mouse hippocampal organotypic brain slices were virally transfected with a genetically encoded glutamate reporter (iGluSnFR under the human synapsin 1 promoter, pAAV.hSynapsin.SF-iGluSnFR.A184S) 1 day post culture using the OpenSpritzer. Five to six days later a brain slice would be removed from the culture insert by cutting a roughly 15-mm diameter circle of the membrane out, with the brain slice located in the centre of the circle. The membrane was then placed in a glass-bottomed Petri dish containing a drop of aCSF, with the brain slice making contact with the glass bottom. At this point, a single *T. crassiceps* larvae (harvested on the same day or 1 day prior) of approximately 1–2 mm in length was inserted under the membrane with a pasteur pipette and carefully maneuvered until it was adjacent to one edge of the slice. A stainless-steel washer of an appropriate size was then placed on top of the membrane such that it surrounded the brain slice and larva, securing them in place. An additional 2–3 ml of carbogen-bubbled aCSF was then added to the Petri dish. The Petri dish was placed in a small incubating chamber (37°C, 5% $CO_2$) in an LSM 880 airyscan confocal microscope (Carl Zeiss, ZEN SP 2 software). Glutamate signalling in the brain area directly adjacent to the larvae was captured with a tile scan of the area every 90 s for 15 min using a 40× water-immersion lens (Zeiss) and a 488-nm laser (Zeiss). Transmitted light was also captured to confirm presence and movement of the larva at the site of interest.

Pharmacological manipulations were performed by bath application of drugs using a perfusion system (Watson-Marlow). Mecamylamine, amiloride, D-AP5, and CNQX were purchased from Tocris. Kynurenic acid and Substance P were acquired from Sigma-Aldrich.

## Data analysis and statistics

Data were graphed and analysed using Matlab, ImageJ, Microsoft Excel, and GraphPad Prism. All data were subjected to a Shapiro–Wilk test to determine whether it was normally distributed. Normally distributed populations were subjected to parametric statistical analyses, whilst skewed data were assessed using non-parametric statistical analyses, these included: Mann–Whitney test; Wilcoxon

ranked pairs test; Kruskal–Wallis one-way ANOVA with post hoc Dunn's multiple comparison test; and Friedman test with post hoc Dunn's multiple comparison test. The confidence interval for all tests was set at 95%.

## Acknowledgements

We would like to acknowledge Dr Philip Fortgens Division of Chemical Pathology, Department of Pathology, University of Cape Town, and National Health Laboratory Service (affiliation when the analyses were done) as well as Dr James T Butler and Dr Roger Melvill of the Epilepsy Unit, Constantiaberg Mediclinic together with two anonymous patients who provided human cortical brain tissue. The research leading to these results has received funding from a Royal Society Newton Advanced Fellowship (NA140170) and a University of Cape Town Start-up Emerging Researcher Award to JVR and grant support from the Blue Brain Project, the National Research Foundation of South Africa, the FLAIR Fellowship Programme (FLR\R1\190829): a partnership between the African Academy of Sciences and the Royal Society funded by the UK Government's Global Challenges Research Fund, a Wellcome Trust Seed Award and a Wellcome Trust International Intermediate Fellowship (222968/Z/21/Z). KAS was supported by a European Commission Marie Sklodowska-Curie Global Fellowship (Grant 657638, WORMTUMORS). CS, UFP, and CPdC were supported by the Federal Ministry of Education and Research of Germany (BMBF), Project title: 'CYSTINET-Africa' (01KA1610, Germany II). The funders had no role in study design, data collection, and analysis, decision to publish, or preparation of the manuscript.

## Additional information

### Funding

| Funder | Grant reference number | Author |
|---|---|---|
| Wellcome Trust | 10.35802/214042 | Joseph V Raimondo |
| Wellcome Trust | 10.35802/222968 | Joseph V Raimondo |
| Royal Society | FLR \R1\190829 | Joseph V Raimondo |
| Royal Society Newton Advanced Fellowship | NA140170 | Joseph V Raimondo |
| The FLAIR Fellowship Programme | FLR\R1\190829 | Joseph V Raimondo |
| European Commission Marie Sklodowska-Curie Global Fellowship | 657638 | Katherine Ann Smith |
| CYSTINET-Africa' | 01KA1610 | Chummy Sikasunge Ulrich Fabien Prodjinotho Clarissa Prazeres da Costa |

The funders had no role in study design, data collection, and interpretation, or the decision to submit the work for publication. For the purpose of Open Access, the authors have applied a CC BY public copyright license to any Author Accepted Manuscript version arising from this submission.

### Author contributions

Anja de Lange, Hayley Tomes, Conceptualization, Data curation, Formal analysis, Investigation, Visualization, Methodology, Writing – original draft, Writing – review and editing; Joshua S Selfe, Formal analysis, Investigation, Visualization; Ulrich Fabien Prodjinotho, Matthijs B Verhoog, Siddhartha Mahanty, Katherine Ann Smith, William Horsnell, Clarissa Prazeres da Costa, Resources, Methodology; Chummy Sikasunge, Resources; Joseph V Raimondo, Conceptualization, Resources, Data curation, Formal analysis, Supervision, Funding acquisition, Investigation, Visualization, Methodology, Writing – original draft, Project administration, Writing – review and editing

## Author ORCIDs

Anja de Lange http://orcid.org/0000-0002-7117-8979
Hayley Tomes http://orcid.org/0000-0002-4495-539X
Katherine Ann Smith https://orcid.org/0000-0001-8150-5702
Joseph V Raimondo https://orcid.org/0000-0002-8266-3128

## Ethics

Ethical approval for use of human tissue was granted by the University of Cape Town Human Research Ethics Committee (HREC 016/2018) according to institutional guidelines. Informed consent, and consent to publish, was obtained from patients prior to temporal lobe surgical resections to treat epilepsy at Mediclinic Constantiaberg Hospital in Cape Town, South Africa.

All animal handling, care, and procedures were carried out in accordance with South African national guidelines (South African National Standard: The care and use of animals for scientific purposes, 2008) and with authorization from the University of Cape Town Animal Ethics Committee (Protocol Nos. AEC 015/015 and AEC 014/035).

Reviewer #1 (Public review): https://doi.org/10.7554/eLife.88174.3.sa1
Reviewer #2 (Public review): https://doi.org/10.7554/eLife.88174.3.sa2
Reviewer #3 (Public review): https://doi.org/10.7554/eLife.88174.3.sa3
Author response https://doi.org/10.7554/eLife.88174.3.sa4

---

# Additional files

## Supplementary files

MDAR checklist

## Data availability

Source data files are available and are linked to each main figure for download.

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
