## [Editor Report · eLife Assessment]

This manuscript addresses infections of the parasite Taenia solium, which causes neurocysticercosis (NCC). NCC is a common parasitic infection that leads to severe neurological problems. It is a major cause of epilepsy, but little is known about how the infection causes epilepsy. The authors used neuronal recordings, imaging of calcium transients in neurons, and glutamate-sensing fluorescent reporters. A strength of the paper is the use of both rodent and human preparations. The results provide **convincing** evidence that the larvae secrete glutamate and this depolarizes neurons. Although it is still uncertain exactly how epilepsy is triggered, the results suggest that glutamate release contributes. Therefore, the paper is a **fundamental** step towards understanding how Taenia solium infection leads to epilepsy.

---

## [Referee Report · Reviewer #1 (Public review)]

In the manuscript, the authors explore the mechanism by which Taenia solium larvae may contribute to human epilepsy. This is extremely important question to address because T. solium is a significant cause of epilepsy and is extremely understudied. Advances in determining how T. solium may contribute to epilepsy could have significant impact on this form of epilepsy. Excitingly, the authors convincingly show that Taenia larvae contain and release glutamate sufficient to depolarize neurons and induce recurrent excitation reminiscent of seizures. They use a combination of cutting-edge tools including electrophysiology, calcium and glutamate imaging, and biochemical approaches to demonstrate this important advance. They also show that this occurs in neurons from both mice and humans. This is relevant for pathophysiology of chronic epilepsy development. This study does not rule out other aspects of T. solium that may also contribute to epilepsy, including immunological aspects, but demonstrates a clear potential role for glutamate.

Strengths:

- The authors examine not only T. solium homogenate, but also excretory/secretory products which suggests glutamate may play a role in multiple aspects of disease progression.

- The authors confirm that the human relevant pathogen also causes neuronal depolarization in human brain tissue

- There is very high clinical relevance. Preventing epileptogenesis/seizures possibly with Glu-R antagonists or by more actively removing glutamate as a second possible treatment approach in addition to/replacing post-infection immune response.

- Effects are consistent across multiple species (rat, mouse, human) and methodological assays (GluSnFR AND current clamp recordings AND Ca imaging)

- High K content (comparable levels to high-K seizure models) of larvae could have also caused depolarization. Adequate experiments to exclude K and other suspected larvae contents (i.e. Substance P).

Weaknesses:

- Acute study is limited to studying depolarization in slices and it is unclear what is necessary/sufficient for *in vivo* seizure generation or epileptogenesis for chronic epilepsy.

- There is likely a significant role of the immune system that is not explored here. This issue is adequately addressed in the discussion, however, and the glutamate data is considered in this context.

Discuss impact:

- Interfering with peri-larval glutamate signaling may hold promise to prevent ictogenesis and chronic epileptogenesis as this is a very understudied cause of epilepsy with unknown mechanistic etiology.

Additional context for interpreting significance:

- High medical need as most common adult onset epilepsy in many parts of the world

---

## [Referee Report · Reviewer #2 (Public review)]

Since neurocysticercosis is associated with epilepsy, the authors wish to establish how cestode larvae affect neurons. The underlying hypothesis is that the larvae may directly excite neurons and thus favor seizure genesis.

To test this hypothesis, the authors collected biological materials from larvae (from either homogenates or excretory/secretory products), and applied them to hippocampal neurons (rats and mice) and human cortical neurons.

This constitutes a major strength of the paper, providing a direct reading of larvae's biological effects. Another strength is the combination of methods, including patch clamp, Ca, and glutamate imaging.

Comments on revised version:

The concerns have been addressed.

---

## [Referee Report · Reviewer #3 (Public review)]

This paper has high significance because it addresses a prevalent parasitic infection of the nervous system, Neurocysticercosis (NCC). The infection is caused by larvae of the parasitic cestode Taenia solium It is a leading cause of epilepsy in adults worldwide

To address the effects of cestode larvae, homogenates and excretory/secretory products of larvae were added to organotypic brain slice cultures of rodents or layer 2/3 of human cortical brain slices from patients with refractory epilepsy.

A self-made pressure ejection system was used to puff larvae homogenate (20 ms puff) onto the soma of patched neurons. The mechanical force could have caused depolarizaton so a vehicle control is critical. On line 150 they appear to have used saline in this regard, and clarification would be good. Were the controls here (and aCSF elsewhere) done with the low Mg2+o aCSF like the larvae homogenates?

They found that neurons depolarized after larvae homogenate exposure and the effect was mediated by glutamate but not nicotinic receptors for acetylcholine (nAChRs), acid-sensing channels or substance P.

They also showed the elevated K+ in the homogenate (~11 mM) could not account for the depolarization. They also confirmed that only small molecules led to the depolarization after filtering out very large molecules. That supports the conclusion that glutamate - which is quite small - could be responsible.

They suggest the effects could underlie seizure generation in NCC.

Using Glutamate-sensing fluorescent reporters they found the larvae contain glutamate and can release it, a strength of the paper.

---

## [Author Response]

The following is the authors’ response to the original reviews.

**Public Reviews:**

**Reviewer #1 (Public Review):**
In the manuscript, the authors explore the mechanism by which Taenia solium larvae may contribute to human epilepsy. This is extremely important question to address because T. solium is a significant cause of epilepsy and is extremely understudied. Advances in determining how T. solium may contribute to epilepsy could have significant impact on this form of epilepsy. Excitingly, the authors convincingly show that Taenia larvae contain and release glutamate sufficient to depolarize neurons and induce recurrent excitation reminiscent of seizures. They use a combination of cutting-edge tools including electrophysiology, calcium and glutamate imaging, and biochemical approaches to demonstrate this important advance. They also show that this occurs in neurons from both mice and humans. This is relevant for pathophysiology of chronic epilepsy development. This study does not rule out other aspects of T. solium that may also contribute to epilepsy, including immunological aspects, but demonstrates a clear potential role for glutamate.Strengths:- The authors examine not only T. solium homogenate, but also excretory/secretory products which suggests glutamate may play a role in multiple aspects of disease progression.- The authors confirm that the human relevant pathogen also causes neuronal depolarization in human brain tissue- There is very high clinical relevance. Preventing epileptogenesis/seizures possibly with Glu-R antagonists or by more actively removing glutamate as a second possible treatment approach in addition to/replacing post-infection immune response.- Effects are consistent across multiple species (rat, mouse, human) and methodological assays (GluSnFR AND current clamp recordings AND Ca imaging)- High K content (comparable levels to high-K seizure models) of larvae could have also caused depolarization. Adequate experiments to exclude K and other suspected larvae contents (i.e. Substance P).Weaknesses:- Acute study is limited to studying depolarization in slices and it is unclear what is necessary/sufficient for *in vivo* seizure generation or epileptogenesis for chronic epilepsy. - There is likely a significant role of the immune system that is not explored here. This issue is adequately addressed in the discussion, however, and the glutamate data is considered in this context.Discuss impact:- Interfering with peri-larval glutamate signaling may hold promise to prevent ictogenesis and chronic epileptogenesis as this is a very understudied cause of epilepsy with unknown mechanistic etiology.Additional context for interpreting significance:- High medical need as most common adult onset epilepsy in many parts of the world

We thank Reviewer 1 for their positive and thorough assessment of our manuscript. We have elected to respond to and address the following aspects from their “Recommendations For The Authors” below:

**Reviewer #1 (Recommendations For The Authors):**
Additional experiments/analysis:- Fig 4a-c: Larva on a slice and not next to it? Negative results maybe because its E/S products are just washed away (assuming submerged recording chamber/conditions)? Experiments and negative results described here do not seem conclusive. Should be discussed at least?

We agree with the reviewer and have added the following sentence to the relevant section of the Results: ‘Our submerged recording setup might have led to swift diffusion or washout of released glutamate, possibly explaining the lack of observable changes.’

Writing & presentation:- Data is not always reported consistently in text and figures, examples:- Results in text are reported varyingly without explanation:- Mean and/or median? SEM or SD and/or IQR? Stat info included in text or not? i.e. lines 130/131 vs. 160/161

Results and data are now presented in a more uniform fashion. We report medians and IQRs, sample size, statistical test result, statistical test used in that order.

- Larval release data interrupts reading flow, lines 246-252 double up results presented in Fig 5F.

This section has now been significantly abbreviated and reads as follows: ‘T. crassiceps larvae released a relatively constant median daily amount of glutamate, ranging from 41.59 – 60.15 ug/20 larvae, which showed no statistically significant difference across days one to six. Similarly, T. crassiceps larvae released a relatively constant median daily amount of aspartate, ranging from 9.431 – 14.18 ug/20 larvae, which showed no statistically significant difference across days one to six.’

- Results in figures are reported in different styles:

Results have now been made uniform, reporting medians and IQRs and: sample size, p test result, statistical test used, figure # reported in that order.

- Fig 6: E/S glu concentration seems to be significantly higher in solium vs crassiceps (about 6fold higher in solium). Should be discussed at least.

Given the small sample size from T. solium (see response below), we do not draw attention to this difference and instead simply make the point that T. solium larvae contain and release glutamate.

- In this context - N=1 may be sufficient for proof of principle (release) but seems too small of a cohort to describe non-constant release of glu over days (Fig 6D). Is initial release on day 1, no release and recovery in the following days reproducible? Is very high glu content of E/S content (15-fold higher in comparison to solium homogenate AND 6-fold higher in comparison to crassiceps homogenate and E/S content). Not sure if Fig 6D is adding relevant information, especially since it is based on n = 1

We agree that a N=1 is only sufficient for proof of principle. However it is worth noting that the measurements still reflect the cumulative release from 20 larvae. Nonetheless, the statement in text has been simplified to say: ‘These results demonstrate that T. solium larvae continually release glutamate and aspartate into their immediate surroundings.’ As this focusses on the point that the larvae release glutamate and aspartate continuously and that we can’t draw conclusions about the variability over days.

Methods:- Human slices, mention cortex - what part, patient data would be interesting. I.e. etiology of epilepsy, epilepsy duration

In the Materials and Methods section “Brain slice preparation” we have now added a table with the requested information.

- For Taenia solium: How were they acquired and used in these experiments?

In the Materials and Methods section “Taenia maintenance and preparation of whole cyst homogenates and E/S products” we describe how Taenia solium larvae were acquired and used.

- Was access resistance monitored? Add exclusion criteria for patch experiments

Figure supplement tables containing the basic properties for each cell recording have been added for each figure and the following statements were added to the electrophysiology section of the Methods: ‘Basic properties of each cell were recorded (supplementary files 1, 2, 3, 4, 6).’ and ‘Cells were excluded from analyses if the Ra was greater than 80 Ω or if the resting membrane potential was above –40 mV.’

- Cannot see any reference to mouse slices in methods? Also, mouse organotypic cultures (for AAV?)? Or only acute slices from mice and organotypic hip cultures from rats? Seems to have been mouse and rat organotypic cultures? But not clear with further clarification in methods.

We have now added the following clarification to the methods: ‘For experiments using calcium and glutamate imaging mouse hippocampal organotypic brain slices were used. For all other experiments rat hippocampal organotypic brain slices were used. A subset of experiments used acute human cortical brain slices and are specified.’

- How long after the wash-in phase was the wash-out phase data collected?

For wash-in recordings drugs were washed in for 8 mins before recordings were made. Drugs were washed out for at least 8 mins before wash-out recordings were made. This information has been added to the Materials and Methods section.

- In general, the M&M section seems to have been written hastily - author's internal remarks "supplier?" are still present.

The M&M section has been thoroughly proofread for errors and internal remarks removed or corrected.

- A little more information on the clinical subjects would be appreciated. I.e. duration of epilepsy? Localization? What cortex? Usual temporal lobe or other regions?

We have now added a table with this information to the Materials and Methods section “Brain slice preparation”.

Minor corrections text/figures:- i.e. 3D,F,H,J show individual data points, thats great, but maybe add mean/median marker (as results are reported like this in text) like in fig 4G,I and others

Figures 3D,F,H & J have been revised to include median and IQR.

- Only one patient mentioned in acknowledgements, but 2 in methods and text

We apologize for this oversight and now acknowledge both patients in the acknowledgements.

- Fig 1 B-F individual puffs are described as increasing - consistent with cellular effects (1st puff depolarizes, 2nd puff elicits 1 AP, 3rd puff elicits AP burst) However, dilution ratio of homogenate or puff concentrations are not mentioned (or potentially longer than 20 ms puffs for 2nd and 3rd stimulus?) in text or figures. Seems to be enough space to indicate in figure as well (i.e. multiple or thicker arrows for subsequent puffs or label with homogenate dilution/concentration in figure).

We state in the results section associated with Fig. 1 that increasing the amount of homogenate delivered was achieved by increasing the pressure applied to the ejection system. We now include this information in the figure legend.

- Figure legend describes 30 ms puff for Ca imaging whereas ephys data (from text) is 20 ms puff. Was Ca imaging performed in acute mouse hippocampal slices (as figure text suggests) or were those organotypic hippocampal cultures from mice?

Ca2+ imaging was performed in mouse hippocampal organotypic brain slice cultures. The figure text for Fig. 1 E states “widefield fluorescence image of neurons in the dentate gyrus of a mouse hippocampal organotypic brain slice culture expressing the genetically encoded Ca2+ reporter GCAMP6s...”

- 11.4 mM K is reported for homogenate in text only. How variable is that? How many n? No SD reported in text and no individual data points reported since this experiment is not represented as a figure.

This has been clarified in the text by adding (N = 1, homogenate prepared from >100 larvae).

- Same results (effect of 11.4 mM K on Vm) described twice in one paragraph, compare lines 126-131 with 131-136.

The repetition has been removed.

- Line 182 - example for consistency: decide IQR or SD/SEM

To improve consistency, we have changed to median and IQR throughout.

- Neuronal recordings are reported as hippocampal pyramidal neurons (i.e. line 222) but some recordings were made from dentate granule cells - please clarify which neurons were recorded in ephys, ca imaging, GluSnFr imaging

For each experiment we describe which type of neurons were recorded from. For rodent recordings these were hippocampal pyramidal neurons except in the case of the Ca2+ imaging example where the widefield recording was over the dentate gyrus subfield.

- Line 309: "should" seems to be an extra word

We have removed the word ‘should’ and made the sentence shorter and clearer. It now reads: ‘Given our finding that cestode larvae contain and release significant quantities of glutamate, it is possible that homeostatic mechanisms for taking up and metabolizing glutamate fail to compensate for larvalderived glutamate in the extracellular space. Therefore, similar glutamate-dependent excitotoxic and epileptogenic processes that occur in stroke, traumatic brain injury and CNS tumors are likely to also occur in NCC.’

**Reviewer #2 (Public Review):**
Since neurocysticercosis is associated with epilepsy, the authors wish to establish how cestode larvae affect neurons. The underlying hypothesis is that the larvae may directly excite neurons and thus favor seizure genesis.To test this hypothesis, the authors collected biological materials from larvae (from either homogenates or excretory/secretory products), and applied them to hippocampal neurons (rats and mice) and human cortical neurons.This constitutes a major strength of the paper, providing a direct reading of larvae's biological effects. Another strength is the combination of methods, including patch clamp, Ca, and glutamate imaging.

We thank the Reviewer 2 for their review of the strength and weaknesses of our manuscript. We respond to the identified weaknesses below.

There are some weaknesses:(1) The main one relates to the statement: "Together, these results indicate that T. crassiceps larvae homogenate results not just in a transient depolarization of cells in the immediate vicinity of application, but can also trigger a wave of excitation that propagates through the brain slice in both space and time. This demonstrates that T. crassiceps homogenate can initiate seizurelike activity under suitable conditions."

The only "evidence" of propagation is an image at two time points. It is one experiment, and there is no quantification. Either increase n's and perform a quantification, or remove such a statement.

We acknowledge that the data is from one experiment, with the intention of demonstrating that it is plausible for intense depolarization of a subset of neurons to result in the initiation and propagation of seizure-like activity to nearby neurons under suitable conditions. However, we agree that it is prudent to remove this statement and have done so.

Likewise, there is no evidence of seizure genesis. A single cell recording is shown. The presence of a seizure-like event should be evaluated with field recordings.

In this experiment the Ca2+ imaging demonstrates activity spreading from the site of the restricted homogenate puff to all surrounding neurons. Furthermore, the whole-cell recoding is typical of a slice wide seizure-like event.

(2) Control puff experiments are lacking for Fig 1. Would puffing ACSF also produce a depolarization, and even firing, as suggested in Fig. 2D? This is needed for at least one species.

We agree and have added this data for the rat and mouse neuron in a new Figure 1-figure supplement 1.

(3) What is the rationale to use a Cs-based solution? Even in the presence of TTX and with blocking K channels, the depolarization may be sufficient to activate Ca channels (LVGs), which would further contribute to the depolarization. Why not perform voltage clamp recordings to directly the current?

The intention of the Cs-based solution was to block K+ channels and reduce the effect of moderately raised K+ in the homogenate to isolate the contribution of other causative agents of depolarization (i.e. glutamate / aspartate). We agree that performing voltage clamp recordings would have been useful for directly recording the currents responsible for depolarization.

(4) Why did you use organotypic slices? Since you wish to model adult epilepsy, it would have been more relevant to use fresh slices from adult rats/mice. At least, discuss the caveat of using a network still in development in vitro.

Recordings were performed 6–14 days post culture, which is equivalent to postnatal Days (P) 12 to 22. Previous work has shown that neurons in the organotypic hippocampal brain slice are relatively mature (Gähwiler et al., 1997). For example they possess mature Cl- homeostasis mechanisms at this point, as evidenced by their hyperpolarizing EGABA (Raimondo et al., 2012).

(5) Please include both the number of slices and number of cells recorded in each condition. This is the standard (the number of cells is not enough).

This has now been added to all relevant sections of the results text.

(6) Please provide a table with the basic properties of cells (Rin, Rs, etc.). This is standard to assess the quality of the recordings.

Tables containing the basic properties for each cell recording have been created for each figure (as Figure supplements) and the following statement was added to the electrophysiology section of the Methods: ‘Basic properties of each cell were recorded (see Figure supplements).’

(7) Please provide a table on patient's profile. This is standard when using human material. Were these TLE cases (and "control" cortex) or epileptogenic cortex?

We have now added a basic table on the patient’s profiles to the Materials and Methods section.

Globally, the authors achieved their aims. They show convincingly that larvae material can depolarize neurons, with glutamate (and aspartate) as the most likely candidates.This is important not only because it provides mechanistic insight but also potential therapeutic targets. The result is impactful, as the authors use quasi-naturalistic conditions, to assess what might happen in the human brain. The experimental design is appropriate to address the question. It can be replicated by any interested person.

We thank the Reviewer 2 for their enthusiastic and constructive assessment of our manuscript. We have elected to respond to and address the following aspects from their “Recommendations For The Authors” below:

**Reviewer #2 (Recommendations For The Authors):**
lines 132 and following are a repetition of those above

These have been removed.

line 151 Fig "2" missing

This has been added.

187, 190 should be E, F not C, D

This has been changed in the text.

481, 482 supplier?

This has been corrected and the correct suppliers described.

**Reviewer #3 (Public Review):**
This paper has high significance because it addresses a prevalent parasitic infection of the nervous system, Neurocysticercosis (NCC). The infection is caused by larvae of the parasitic cestode Taenia solium It is a leading cause of epilepsy in adults worldwideTo address the effects of cestode larvae, homogenates and excretory/secretory products of larvae were added to organotypic brain slice cultures of rodents or layer 2/3 of human cortical brain slices from patients with refractory epilepsy.

We thank Reviewer 3 for their helpful comments and suggestions for improvement which we address below.

A self-made pressure ejection system was used to puff larvae homogenate (20 ms puff) onto the soma of patched neurons. The mechanical force could have caused depolarizaton so a vehicle control is critical. On line 150 they appear to have used saline in this regard, and clarification would be good. Were the controls here (and aCSF elsewhere) done with the low Mg2+o aCSF like the larvae homogenates?

We agree and have added examples where aCSF alone was pressure ejected onto the same rat and mouse neurons in a new Figure 1-figure supplement 1. In Figure 1, the same aCSF as that was used to bathe the slices was used. In Figure 2D-G, either PBS (which larval homogenates were prepared in) or growth medium (which contain larval E/S products) were used as comparative controls.

They found that neurons depolarized after larvae homogenate exposure and the effect was mediated by glutamate but not nicotinic receptors for acetylcholine (nAChRs), acid-sensing channels or substance P. To address nAChRs, they used 10uM mecamyline, and for ASICs 2mM amiloride which seems like a high concentration. Could the concentrations be confirmed for their selectivity?

We did not independently verify the selectivity of the antagonist concentrations used in our study. However, the persistence of depolarizations despite the use of high concentrations of mecamylamine (10 μM) and amiloride (2 mM) provides strong evidence that neither nAChRs nor ASICs are primarily responsible for mediating these responses. The high concentrations used, while potentially raising concerns about specificity, actually strengthen our conclusion that these receptor types are not involved in the observed effect.

Glutamate receptor antagonists, used in combination, were 10uM CNQX, 50uM DAP5, and 2mM kynurenic acid. These concentrations are twice what most use. Please discuss.

We intentionally used higher-than-typical concentrations of glutamate receptor antagonists in our experimental design. Our rationale for this approach was to ensure maximal blockade of glutamate receptors, thereby minimizing the possibility of residual receptor activity confounding our results.

Also, it would be very interesting to know if the glutamate receptor is AMPA, Kainic acid, or NMDA. Were metabotropic antagonists ever tested? That would be logical because CNQX/DAPR/Kynurenic acid did not block all of the depolarization.

We appreciate the reviewer's interest in the specific glutamate receptor subtypes involved in our study. Our research primarily focused on ionotropic glutamate receptors as a group, without differentiating the individual contributions of AMPA, Kainate, and NMDA receptors. This approach, while broad, allowed us to establish the involvement of glutamatergic signalling in the observed effects. We acknowledge that we did not investigate metabotropic glutamate receptors in this study. Importantly, we demonstrate later in our manuscript that the larval products contain both glutamate and aspartate. Therefore the precise nature of the glutamate-dependent depolarization observed using a particular experimental preparation would depend on the specific types of neurons exposed to the homogenate and the expression profile of different glutamate receptor subtypes on these neurons.

They also showed the elevated K+ in the homogenate (~11 mM) could not account for the depolarization. However, the experiment with K+ was not done in a low Mg2+o buffer (Or was it -please clarify).

The experiment where 11.39 mM K+ as well as the experiment with T. crass. Homogenate with a cesium internal and added TTX were all done in standard 2 mM Mg2+ containing aCSF.

They also confirmed that only small molecules led to the depolarization after filtering out very large molecules. That supports the conclusion that glutamate - which is quite small - could be responsible. It is logical to test substance P because the Intro points out prior work links the larvae and seizures by inflammation and implicates substance P. However, why focus on nAChRs and ASIC?

These were chosen as they are ionotropic receptors which mediate depolarization and hence could conceivably be responsible for the homogenate-induced depolarization we observed.

The depolarizations caused seizure-like events in slices. The slices were exposed to a proconvulant buffer though- low Mg2+o. This buffer can cause spontaneous seizure-like events so it is important to know what the buffer did alone.

We agree that a low M2+ buffer solution can elicit seizure-like events in organotypic slices alone. However, the timing of the onset of the seizure-like event in the example presented in Figure 1 strongly suggests that it was triggered by the T. crass homogenate puff. Nonetheless, on the suggestion of the other reviewers we have reduced emphasis on our experimental evidence for the ability of T. crass. homogenate to illicit seizure-like events.

They suggest the effects could underlie seizure generation in NCC. However, there is only one event that is seizure-like in the paper and it is just an inset. Were others similar? How frequency were they? How long?

Please see the response above as well as our response to Reviewer 1 who raised a similar concern.

Using Glutamate-sensing fluorescent reporters they found the larvae contain glutamate and can release it, a strength of the paper.Fig. 4. Could an inset be added to show the effects are very fast? That would support an effect of glutamate.

We have not added an inset. However, given the scale bar (500 ms) for the trace provided, the response is very fast.

Why is aspartate relatively weak and glutamate relatively effective as an agonist?

Glutamate generally has a higher affinity for glutamate receptors compared to aspartate. This is particularly true for AMPA and kainate receptors, where glutamate is the primary endogenous agonist. Similarly iGluSnFR has a higher sensitivity for glutamate over aspartate (Marvin et al., 2013).

Could some of the variability in Fig 4G be due to choice of different cell types? That would be consistent with Fig 5B where only a fraction of cells in the culture showed a response to the larvae nearby.

Whilst differences in cell types could contribute to the variability in Fig 4G, all the responses were recorded from hippocampal pyramidal neurons and hence it is more likely that the variability is a function of other sources of variation including differences in iGluSnFR expression, depth of the cell imaged, the proximity of the puffer pipette etc. In Fig. 5B we think the lack of response may be due to the fact that any released glutamate by the live larvae was not able reach the iGluSnFR neurons at sufficient concentrations due to the nature of our submerged recording setup. We have added the following sentence to the results. ‘Our submerged recording setup might have led to swift diffusion or washout of released glutamate, possibly explaining the lack of observable changes.’

On what basis was the ROI drawn in Fig. 5B.

The ROI drawn in Fig. 5B was selected to include all iGluSnFR expressing neurons in the brain slice. which were captured in the field of view.

Also in 5B, I don't see anything in the transmitted image. What should be seen exactly?

We agree that it is difficult to resolve much in the transmitted image. However, both the brain slice on the left as well as a T. crass. larva on the right is visible and outlined with a green or orange dashed line respectively.

Human brain slices were from temporal cortex of patients with refractory epilepsy. Was the temporal cortex devoid of pathology and EEG abnormalities? This area may be quite involved in the epilepsy because refractory epilepsy that goes to surgery is often temporal lobe epilepsy. Please discuss the limitations of studying the temporal cortex of humans with epilepsy since it may be more susceptible to depolarizations of many kinds, not just larvae.

We acknowledge the important limitations of using temporal cortex tissue from patients with refractory epilepsy. While we aimed to use visually normal tissue, we recognize that the tissue may have underlying pathology or functional abnormalities not visible to the naked eye. It may also be more susceptible to induced depolarizations due to epilepsy-related changes in neuronal excitability. Despite these limitations, we believe our human tissue data still provides valuable data that the larval homogenates can induce depolarization in human as well as rodent neurons.

Please discuss the limitations of the cultures - they are from very young animals and cultured for 6-14 days.

We acknowledge the potential limitations of our experimental model using organotypic hippocampal slice cultures from young animals. The use of relatively immature tissue may not fully represent the adult nervous system due to developmental differences in receptor expression, synaptic connections, and network properties. The 6-14 day culture period, while allowing some maturation, may induce changes that differ from the in vivo environment, including alterations in cellular physiology and network reorganization. Despite these limitations, this model provides a valuable balance between preserved local circuitry and experimental accessibility. Future studies comparing results with acute adult slices and in vivo models would be beneficial to validate and extend our findings.

References:

Gähwiler, B.H. et al. (1997) ‘Organotypic slice cultures: a technique has come of age.’, Trends in neurosciences, 20(10), pp. 471–7.

Marvin, J.S. et al. (2013) ‘An optimized fluorescent probe for visualizing glutamate neurotransmission.’, Nature methods, 10(2), pp. 162–70. Available at: https://doi.org/10.1038/nmeth.2333.

Raimondo, J.V. et al. (2012) ‘Optogenetic silencing strategies differ in their effects on inhibitory synaptic transmission.’, Nat. Neurosci., 15(8), pp. 1102–4. Available at: https://doi.org/10.1038/nn.3143.